# WSM: Decay-Free Learning Rate Schedule via Checkpoint Merging for LLM Pre-training

**Changxin Tian**[1][*], **Jiapeng Wang**[1,2,3][*][‡], **Qian Zhao**[1], **Kunlong Chen**[1], **Jia Liu**[1],
**Ziqi Liu**[1], **Jiaxin Mao**[2,3], **Wayne Xin Zhao**[2,3][†], **Zhiqiang Zhang**[1][†], **Jun Zhou**[1]

[1]Ling Team, Ant Group  [2]Gaoling School of Artificial Intelligence, Renmin University of China
[3]Beijing Key Laboratory of Research on Large Models and Intelligent Governance
tianchangxin.tcx@antgroup.com, wangjp1010@ruc.edu.cn

## Abstract

Recent advances in learning rate (LR) scheduling have demonstrated the effectiveness of decay-free approaches that eliminate the traditional decay phase while maintaining competitive performance. Model merging techniques have emerged as particularly promising solutions in this domain. We present Warmup-Stable and Merge (WSM), a general framework that establishes a formal connection between learning rate decay and model merging. WSM provides a unified theoretical foundation for emulating various decay strategies—including cosine decay, linear decay and inverse square root decay—as principled model averaging schemes, while remaining fully compatible with diverse optimization methods. Through extensive experiments, we identify merge duration—the training window for checkpoint aggregation—as the most critical factor influencing model performance, surpassing the importance of both checkpoint interval and merge quantity. With the high-quality annealing data, our framework consistently outperforms the widely-adopted Warmup-Stable-Decay (WSD) approach across multiple benchmarks, achieving significant improvements of +3.5% on MATH, +2.9% on HumanEval, and +5.5% on MMLU-Pro. The performance advantages extend to supervised fine-tuning scenarios, highlighting WSM's potential for long-term model refinement.

## 1 Introduction

In large language model (LLM) pre-training, learning rate (LR) scheduling plays a pivotal role, critically impacting training stability, convergence speed, and final model performance (Jin et al., 2023; Gotmare et al., 2019). Conventional LR schedules dynamically adjust the LR based on training

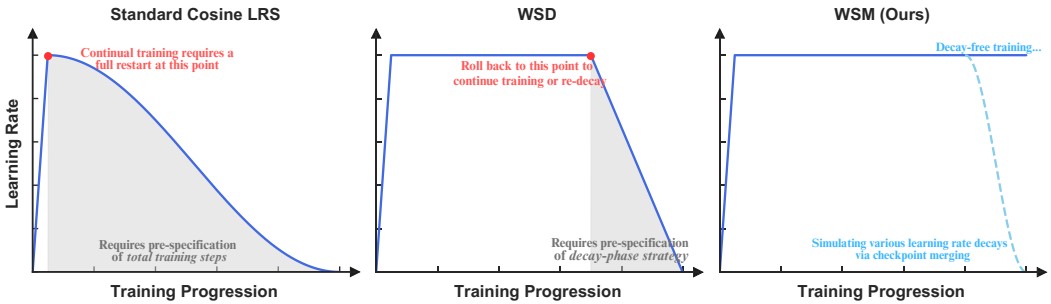

Figure 1: Comparison between our proposed WSM (Warmup-Stable and Merge) and mainstream learning rate scheduling strategies, Cosine (Loshchilov & Hutter, 2016) and WSD (Hu et al., 2024).

---

[*]Equal contribution. [‡]Contribution during internship at Ant Group.
[†]Corresponding authors: Wayne Xin Zhao (batmanfly@gmail.com) and Zhiqiang Zhang (lingyao.zzq@antgroup.com).

progress, with the cosine decay schedule emerging as a widely-adopted approach (Loshchilov & Hutter, 2016; Kaplan et al., 2020; Hoffmann et al., 2022). This method features an initial warm-up phase followed by cosine-based decay, both constrained by a *predetermined* total training duration. Consequently, any extension of the training process, such as incorporating new data, necessitates a complete restart from the beginning to recalibrate the entire decay curve.

To address this inflexibility, the Warmup-Stable-Decay (WSD) strategy (Hu et al., 2024) inserts a *stable training* phase with a constant LR between the warmup and decay phases. This schedule provides two key advantages: (1) flexible initiation of the decay phase independent of total step count, and (2) complete decoupling from fixed training durations. The WSD approach has demonstrated significant effectiveness, evidenced by its adoption in several recent LLMs including DeepSeek-V3 (DeepSeek-AI et al., 2024) and ERNIE 4.5 (ERNIE-Team, 2025). However, WSD introduces new scheduling requirements: researchers must manually decide when to initiate the decay, how many tokens to allocate for it, and which decay function (e.g., cosine, linear) to employ. Moreover, if training needs to be extended after the decay has commenced, one must roll back the training to the state preceding the decay phase and re-design the decay strategy. This dependency on a manually configured schedule counteracts the goal of a fully autonomous and continuous training process.

To minimize scheduling complexity, recent research has investigated alternative approaches that completely eliminate the decay phase from LR schedules (Defazio et al., 2024; Song et al., 2025; Zhang et al., 2025). A prominent direction in this field explores weight averaging (a.k.a., model merging) techniques. Empirical results demonstrate that simply maintaining a constant LR combined with standard weight averaging strategies—such as exponentially weighted averaging (EWA)—can achieve performance competitive with WSD-based schedules (Li et al., 2025). Building upon this line of research, we make three key extensions:

(1) We present *Warmup-Stable and Merge (WSM)*, a simple yet general framework for LR scheduling. By formalizing the connection between LR decay and checkpoint merging, we demonstrate that WSM can be instantiated to emulate various decay strategies—including cosine decay, linear decay and inverse square root (1-*sqrt*) decay. Our framework provides a principled approach to convert any LR decay method into a theoretically approximate model averaging implementation. This contrasts with prior work (Defazio et al., 2024; Song et al., 2025; Li et al., 2025), which has largely focused on analysis of specific averaging strategies and their optimization properties. Notably, our framework is optimizer-agnostic, enabling seamless integration with various optimization algorithms (e.g., SGD, Adam) without requiring modifications on the underlying training pipeline.

(2) Our extensive experiments systematically investigate key factors for instantiating this framework, including merge methods, merge frequency, duration, granularity, and the compatibility of merging and decay strategies. Our findings reveal that, aside from the introduction of high-quality annealing data during the decay (i.e., merge) phase, merge duration—the training period covered by the merged checkpoints—emerges as the most critical factor influencing model performance, with significantly greater impact than both the checkpoint interval and the number of merged models.

(3) The proposed decay-free LR schedule delivers substantial performance gains. Extensive empirical evaluation shows consistent improvements across multiple benchmarks: +3.5% on MATH, +2.9% on HumanEval, and +5.5% on MMLU-Pro—a significant advance over the WSD method, while prior work often only matches WSD's performance. Moreover, these benefits naturally extend to the post-training stage, demonstrating our method's potential for sustained improvements in long-term model refinement.

The proposed WSM framework presents a promising direction for developing effective decay-free LR schedules. Our results demonstrate consistent performance advantages across both pre-training and fine-tuning stages. Notably, WSM enables the implementation of sophisticated decay-like methods through a simple and stable optimization approach, combining the benefits of strategic scheduling with robust training dynamics.

## 2 PRELIMINARY

Typically, existing mainstream LR schedulers consist of two phases: an initial warm-up followed by decay. During the warm-up, the LR increases linearly from a small value to a peak value, $lr_{peak}$, over $T_{warmup}$ steps. This helps stabilize the optimization process in the early stages of training.

Following the warm-up, the LR gradually decreases according to a predefined function, such as cosine, linear, or inverse square root decay. These schedules share a critical limitation: they require the total number of training tokens (or steps), $T_{max}$, to be known in advance. For instance, the prevalent cosine LR schedule is formulated as:

$$lr(t) = \begin{cases} lr_{peak} \cdot \frac{t}{T_{warmup}} & \text{if } t < T_{warmup} \\ \frac{1}{2} lr_{peak} \left( 1 + \cos \left( \frac{\pi(t - T_{warmup})}{T_{max} - T_{warmup}} \right) \right) & \text{if } t \geq T_{warmup} \end{cases}$$

The Warmup-Stable-Decay (WSD) LR schedule introduces a *stable* phase between warm-up and decay, maintaining a constant LR at its peak ($lr_{peak}$), which is defined as:

$$lr(t) = \begin{cases} lr_{peak} \cdot \frac{t}{T_{warmup}} & \text{if } t < T_{warmup} \\ lr_{peak} & \text{if } T_{warmup} \leq t < T_{decay\_start} \\ \text{decay\_function}(t) & \text{if } T_{decay\_start} \leq t \leq T_{max} \end{cases}$$

As shown, WSD facilitates flexible experimentation by supporting multiple decay attempts from the endpoint of the stable phase—without requiring a reset to the initial state. Despite this flexibility, it still requires predefined decay-phase settings—such as the decay start step ($T_{decay\_start}$), decay function (decay_function), and total training steps ($T_{max}$).

## 3 THE PROPOSED METHODOLOGY

In this section, we first establish the theoretical connection between checkpoint merging and LR decay, formalize our proposed WSM (Warmup-Stable and Merge) schedule, and compare it with the widely used WSD schedule.

### 3.1 THEORETICAL CONNECTION BETWEEN LR DECAY AND CHECKPOINT MERGING

The core idea of checkpoint merging in this work is to take an ordered list of checkpoints, $[\theta_n, \theta_{n+1}, \ldots, \theta_{n+k}]$, and apply a merge function to generate a single model $\hat{\theta}_{n+k}$. Here, $\theta_i \in \mathbb{R}^d$ represents the model's parameter vector at the $i$-th training iteration. The most general form is a weighted average of the checkpoints:

$$\hat{\theta}_{n+k} = \sum_{j=0}^{k} c_j \theta_{n+j} \tag{1}$$

where $\{c_j\}$ are non-negative weights that sum to one, i.e., $\sum_{j=0}^{k} c_j = 1$. This formulation obscures a deeper connection to the training dynamics. We can reveal this connection by expressing each checkpoint in terms of an initial checkpoint $\theta_n$ and the subsequent gradient updates. For simplicity, we assume the updates between checkpoints at different time steps are independent and ignore optimizer states. Let $g_i$ be the gradient update vector (including the LR) at step $i$, such that the model updates as $\theta_{i+1} = \theta_i - g_i$. Thus, an intermediate checkpoint $\theta_{n+j}$ can be expressed as the sum of an initial state $\theta_n$ and the sequence of negative gradient updates that followed:

$$\theta_{n+j} = \theta_n - \sum_{l=1}^{j} g_{n+l-1} \tag{2}$$

Furthermore, we can substitute Eq. 2 into the general merging formula (Eq. 1) and then rearrange the double summation by changing its order. A gradient update $g_{n+i-1}$ is included in the sum for all involved checkpoints $\theta_{n+j}$ where $j \geq i$.

$$\hat{\theta}_{n+k} = \sum_{j=0}^{k} c_j \left( \theta_n - \sum_{l=1}^{j} g_{n+l-1} \right) = \theta_n - \sum_{i=1}^{k} \left( \sum_{j=i}^{k} c_j \right) g_{n+i-1} \tag{3}$$

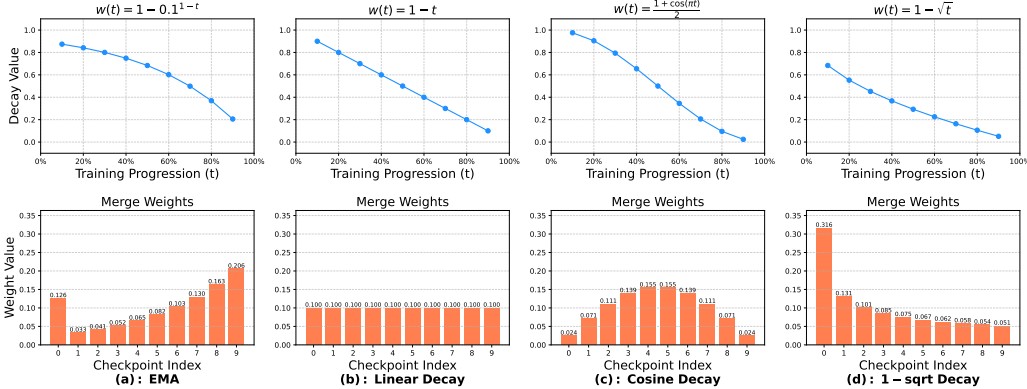

Figure 2: Visualization of checkpoint merging weight distributions and their corresponding decay functions over a span of 10 checkpoints (larger checkpoint indices denote more recent checkpoints). (a) Exponential Moving Average (EMA) weights, exhibiting convex decay characteristics; (b) Uniform averaging weights, demonstrating linear decay behavior; (c) and (d) Our theoretically derived weights, designed to approximate cosine and 1-*sqrt* decay patterns. Note: Decay curves are rendered smoothly for clarity, though the underlying weights are applied discretely at each checkpoint.

This shows that a weight coefficient of $\sum_{j=i}^{k} c_j$ is applied to the gradient update $g_{n+i-1}$ from step $n + i - 1$. By defining a new set of weights for the gradient updates, $w_i = \sum_{j=i}^{k} c_j$, we arrive at the final equivalent expression for checkpoint merging:

$$\hat{\theta}_{n+k} = \theta_n - \sum_{i=1}^{k} w_i \cdot g_{n+i-1} \tag{4}$$

This equation shows that merging checkpoints with weights $c_j$ is equivalent to applying a synthetic decay schedule defined by weights $w_i$ to the gradients accumulated after the base checkpoint $\theta_n$, where a mapping exists between the merge weights $c_j$ and the effective learning rates $w_i$. We therefore propose Theorem 3.1, which can approximate monotonically decreasing decay curve. Figure 2 illustrates various checkpoint merging weights and their corresponding decay curves. Additional details are provided in Appendix B.

**Theorem 3.1** (Checkpoint Weight Derivation from Gradient Decay Schedule). *Given a desired sequence of gradient decay coefficients $\{w_i\}_{i=1}^{k}$ that is monotonically non-increasing and bounded, $1 \geq w_1 \geq w_2 \geq \cdots \geq w_k \geq 0$, the corresponding non-negative checkpoint weights $\{c_j\}_{j=0}^{k}$ required to satisfy Eq. 4 are uniquely determined by:*

$$\begin{cases} c_k = w_k \\ c_j = w_j - w_{j+1}, & \text{for } j \in [1, k-1] \\ c_0 = 1 - \sum_{j=1}^{k} c_j = 1 - w_1 \end{cases} \tag{5}$$

## 3.2 EMULATING LR DECAY THROUGH CHECKPOINT MERGING

The central hypothesis of WSM is that *the optimization benefits of LR decay can be decoupled from the live training process and instead be effectively achieved through the merging of model checkpoints* (Kingma & Ba, 2015; DeepSeek-AI et al., 2024; Li et al., 2025). The WSM simplifies the LR schedule by completely eliminating the decay phase:

$$lr(t) = \begin{cases} lr_{peak} \cdot \frac{t}{T_{warmup}} & \text{if } t < T_{warmup} \\ lr_{peak} & \text{if } t \geq T_{warmup} \end{cases}$$

The WSM pipeline, detailed in Algorithm 1, operates in two primary phases. It begins with a standard warmup phase, where the learning rate linearly increases to its peak value, $lr_{peak}$. Following this, the process enters the main stable training phase, where the learning rate is held constant.

After a specified step $T_{switch}$, the model can transition from the general pre-training data $D$ to a smaller, high-quality annealing dataset $D_{anneal}$, allowing the "annealing" to focus on a curated data mixture. Throughout this stable phase, checkpoints are periodically saved. Concurrently, an asynchronous merging process continuously fetches the last $n$ checkpoints from storage and combines them into model $W_{merged}$. Specifically, for the $\text{Merge}(\cdot)$ operation, we can select various decay strategies to emulate (e.g., the decay curve shape and minimum LR), calculate the corresponding gradient decay coefficients $\{w_i\}$, and then derive the checkpoint merging weights $\{c_i\}$ based on Theorem 3.1. This merged checkpoint, which emulates the effect of a decay schedule, provides a robust, annealed model without ever altering the live learning rate.

## 4 EXPERIMENT

Next, we present the empirical evaluation of our proposed WSM to validate its effectiveness.

### 4.1 EXPERIMENT SETUP

The model we used for the main experiment is a standard MoE model with a total of 16.3 billion parameters and 1.4 billion active parameters. We utilized the AdamW optimizer (Loshchilov & Hutter, 2019), and the hyperparameters are set to $\beta_1 = 0.9$ and $\beta_2 = 0.95$, with 0.1 weight decay applied. Through preliminary scaling laws experiments, we set the peak LR and batch size to 4.78e-4 and 2048, respectively. Comprehensive details regarding our model architecture, specific training parameters, dataset composition, and evaluation protocols are provided in Appendix A.

We begin with a checkpoint pretrained on 10.2 trillion tokens from the *pretraining dataset* using a stable, constant LR. Then, we continue training for an additional 400B tokens on a specialized *high-quality annealing dataset*, following two distinct strategies to evaluate their relative effectiveness: (1) We apply a conventional LR decay schedule to the model. This branch serves as our baseline, representing the standard Warmup-Stable-Decay (WSD) methodology. Note that we employ the standard re-warm-up strategy for continual pre-training as in a standard WSD setting. (2) We continue training with the same constant LR. The final model is then produced by merging the checkpoints saved during this stable phase. This branch represents our proposed WSM schedule. Unless otherwise specified, we save a checkpoint every 25B tokens and use mean averaging to merge the most recent checkpoints, which corresponds to a linear decay LR schedule.

### 4.2 OVERALL PERFORMANCE OF WSM SCHEDULE

**Effectiveness in pre-training** We evaluate our WSM schedule against the baseline WSD using three series of checkpoints: (1) those obtained using the standard LR decay schedule in WSD, (2) checkpoints from the last stable phase of WSM before merging, and (3) our final merged checkpoints from the WSM method using various merge durations (window sizes). The category-wise average results are summarized in Figure 3 and Table 1, and scores for each benchmark are provided in Appendix J.

Our first and most significant finding is that the WSM method consistently outperforms WSD across the majority of tasks considered. On average, WSM achieves performance improvements across all benchmark categories. Notably, when comparing the best-performing checkpoint from each strategy, WSM improves upon WSD by an average of 1.3 points. We observe remarkable improvements of up to 2.7 points on MATH, 2.4 points on HumanEval, 2.1 points on MMLU-Pro. These results provide compelling evidence that replacing the LR decay phase used in WSD with the checkpoint merging strategy of WSM is not only a feasible alternative but also a more effective approach for enhancing the diverse capabilities of the final pretrained model.

**Long-term effectiveness for post-training** To assess the long-term impact of WSM, we extended our evaluation to the post-training phase. We apply supervised fine-tuning on checkpoints generated by the WSM and WSD under identical settings for 5 epochs. Results in Table 2 show that this advantage persists beyond the post-training phase.

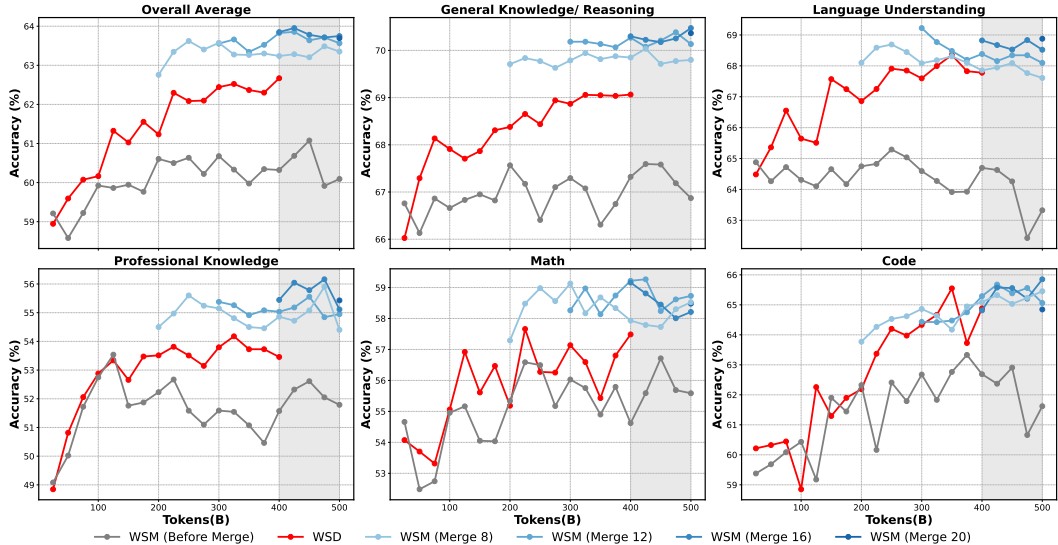

Figure 3: Comprehensive performance comparison (overall and by category) between our WSM schedule (via checkpoint merging, blue line) and standard WSD scheduling (via LR decay, red line). Notably, WSD requires a predefined decay schedule (e.g., over 400B tokens in this study), whereas WSM eliminates this constraint. This flexibility enables seamless training continuation (gray regions) and allows WSM to approximate various decay curves.

Table 1: Base model performance comparison. Results are reported based on the checkpoint with the highest average benchmark score.

|  |  | General Knowledge | Language Modeling | Math | Code | Professional Knowledge | Overall Average |
|---|---|---|---|---|---|---|---|
| Base Model | WSD | 69.06 | 67.78 | 57.49 | 64.88 | 53.46 | 62.67 |
|  | WSM | **70.22** | **68.67** | **58.81** | **65.58** | **56.04** | **63.95** |
|  | Improv. | +1.68% | +1.31% | +2.30% | +1.08% | +4.83% | +2.04% |

Table 2: Instruct model performance comparison. Results are reported based on the epoch with the highest average benchmark score.

|  |  | Language | Knowledge | Math | Code | Reason | Agent | Overall Average |
|---|---|---|---|---|---|---|---|---|
| Inst Model | WSD | 81.12 | 60.00 | 61.43 | **58.23** | 63.21 | 68.16 | 62.90 |
|  | WSM | **84.78** | **61.73** | **62.28** | 57.95 | **64.94** | **69.33** | **64.07** |
|  | Improv. | +4.51% | +2.88% | +1.38% | -0.48% | +2.74% | +1.72% | +1.86% |

## 4.3 EMPIRICAL ANALYSIS OF WSM SCHEDULE

Next, we conduct a comprehensive empirical study to dissect the WSM schedule. We aim to understand the key factors influencing its performance, its robustness across different training scenarios, its interaction with conventional decay schedule, and its broader implications on model dynamics.

### 4.3.1 ROBUSTNESS ACROSS PRE-TRAINING PROCESS

Beyond applying WSM as a final step on a high-quality dataset, we also evaluated its utility and robustness throughout the entire pre-training lifecycle. To achieve this, we conducted a comparative analysis at various intermediate stages of a long training run, comparing the performance of a model produced by our computationally-frugal WSM merging against one produced by initiating a full,

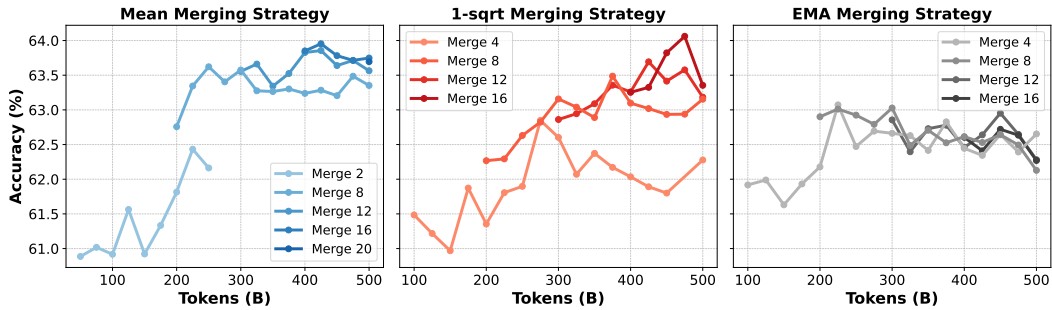

Figure 4: Merge duration analysis across algorithms. "Merge 4" indicates merging the most recent 4 checkpoints, with each checkpoint saved at 25B-token intervals.

resource-intensive LR decay. As shown in Figure 5a, although the performance gains over WSD are not as significant as when switching to high-quality data, we find that the performance of models generated via WSM merging consistently and closely mirrors the results of a true LR anneal. In the figure, the gray line represents the base model trained with a constant LR. The blue lines show the performance of WSM models, which were created by mean-merging four checkpoints from within the preceding 100B-token merge duration. The red lines represent full 100B-token decay runs initiated at the 2T, 4T, 6T, 8T, and 10T token milestones. This result demonstrates that the incorporation of high-quality annealing data is the key factor enabling WSM to outperform WSD, while simultaneously establishing WSM as a reliable, high-fidelity proxy for estimating a model's post-anneal potential at any point during training. Consequently, it can provide effective assessment throughout the pre-training phase, eliminating the need to launch multiple, expensive decay runs to determine the model's ultimate performance.

### 4.3.2 IMPACT OF MERGING ALGORITHM

As derived in Section 3.1, the checkpoint merging process can be viewed as an approximation of a LR decay schedule, where the weighting scheme of the merge is analogous to the functional form of the decay curve. For instance, a simple mean average is analogous to a linear decay curve. An Exponential Moving Average (EMA) would correspond to a convex exponential decay curve. Existing works (Hägele et al., 2024) and our prior experiments (details are provided in Appendix F) with the WSD schedule revealed a performance hierarchy among decay curves: concave schedules (e.g., inverse square root) and linear schedules outperform convex schedules. Building on these findings, we hypothesize that a merging algorithm designed to approximate decay curves that have been proven effective in WSD scheduling will similarly yield better results.

Based on Theorem 3.1, we experimentally compare three merging algorithms: one using our theorem-generated weights to approximate 1-*sqrt* decay, another using simple mean averaging, and a third using EMA. Our experimental results in Table 3 validate this hypothesis. While the merge method outperforms decay, we observe a distinct performance ranking: the 1-*sqrt* merge approach outperforms Mean, and both are markedly better than EMA. Crucially, this ranking (1-*sqrt* > Mean > EMA) is identical to the hierarchy observed in standard LR decay schedules. This consistency demonstrates that WSM reliably reproduces the relative effectiveness of different decay schedules, confirming that checkpoint merging serves as a principled and effective simulation of the LR decay process.

### 4.3.3 IMPACT OF MERGING DURATION AND GRANULARITY

We further investigate in detail the impact of different merge durations (window sizes) on various algorithms. First, different merge durations essentially correspond to decaying with varying amounts of data. As shown in Figure 4, when comparing the best-performing checkpoints across the entire merging trajectory for both mean and 1-*sqrt* merging algorithms, larger merging windows tend to yield better results. However, this advantage gradually diminishes as the window size increases. This observation aligns with previous practical LR decay experiments, where simply increasing the amount of annealing data shows diminishing returns and may eventually fail to provide further

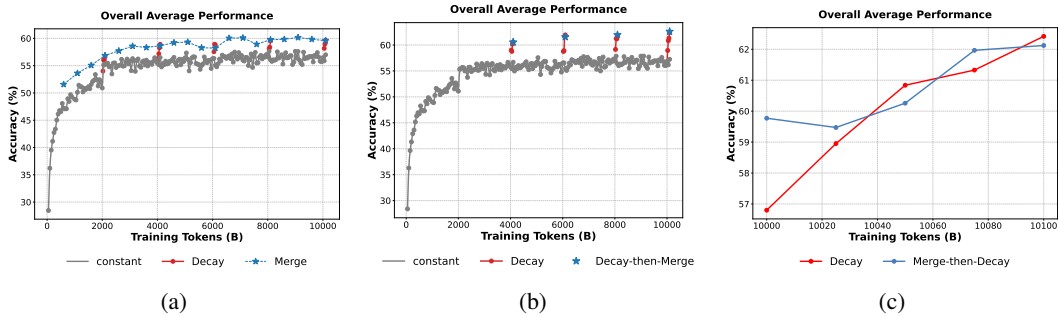

Figure 5: Interplay of checkpoint merging and LR decay. (a) A comparison of checkpoint merging (WSM) and true LR decay (WSD) across a long pre-training run. (b) Investigating the effect of applying merging within a decay phase (Decay-then-Merge). (c) Investigating the effect of applying LR decay after checkpoint merging (Merge-then-Decay).

improvement. For EMA merging, its performance is significantly inferior to other algorithms and shows no clear variation with merge durations. This may indicate that EMA is not an effective merging algorithm (also suggesting that the convex nature may not represent an optimal decay curve).

Then we further investigate the granularity of checkpoint merging, i.e., the interval between saved checkpoints used for merging. Finer-grained merging represents a more precise approximation of the true decay curve. The results in Table 4 show that finer-grained merging tends to achieve better performance. However, frequent checkpoint saving imposes storage overhead, requiring careful trade-off considerations.

Table 3: Impact of merging algorithm.

|  |  | General Knowledge | Language Modeling | Math | Code | Professional Knowledge | Overall Average |
|---|---|---|---|---|---|---|---|
| Decay | 1-*sqrt* | 69.06 | 67.78 | 57.49 | 64.88 | 53.46 | 62.67 |
| Merging | EMA | 69.05 | 67.64 | 58.81 | 64.19 | 54.44 | 63.01 |
|  | Mean | 70.22 | **68.67** | 58.81 | 65.58 | **56.04** | 63.95 |
|  | 1-*sqrt* | **70.27** | 68.26 | **59.65** | **65.70** | 55.42 | **64.06** |

Table 4: Comparison of different saving/merging intervals within an 80B-token merge duration. For example, (5B,16) indicates that saving every 5B tokens while merging the latest 16 checkpoints.

| Merging Granularity | General Knowledge | Language Modeling | Math | Code | Professional Knowledge | Overall Average |
|---|---|---|---|---|---|---|
| (5B,16) | **69.46** | 79.95 | 57.07 | **64.68** | 53.82 | 63.63 |
| (10B,8) | 69.23 | 80.00 | **57.94** | 64.39 | 53.78 | **63.78** |
| (20B,4) | 69.29 | **80.29** | 56.79 | 63.87 | **54.19** | 63.36 |
| (40B,2) | 68.47 | 79.83 | 56.57 | 63.59 | 52.20 | 62.77 |
| (80B,1) | 67.47 | 64.98 | 55.07 | 61.69 | 51.61 | 60.33 |

### 4.3.4 ON THE COMPATIBILITY OF MERGE AND DECAY

Given that checkpoint merging effectively simulates LR decay, a natural question arises: can merging and decay be combined to achieve synergistic performance gains? We investigate this by testing two hybrid approaches. (1) Decay-then-Merge: We first apply a standard decay schedule and then merge checkpoints selected from within the decay phase. (2) Merge-then-Decay: We further apply a decay schedule to the resulting merged model. As shown in Figures 5b and 5c, the hybrid approach failed to yield any improvement in either configuration, although the Merge-then-Decay model showed better performance at the beginning of its training. For the Decay-then-Merge experiment (Figure 5b), the blue stars represent the results of merging checkpoints selected along the

decay trajectories (red lines), which were initiated at various pre-training milestones (4T, 6T, 8T, 10T). For the Merge-then-Decay experiment (Figure 5c), we compare a decay run initiated from a WSM model (blue line)—created by merging four checkpoints from the 9.8T to 10T token interval—against a standard decay baseline initiated from a single 10T checkpoint (red line). These results suggest that checkpoint merging and LR decay are not complementary but rather alternative pathways to a similar optimization objective.

### 4.3.5 IMPACT ON MoE LOAD BALANCING

Table 5: Impact on MoE load balancing. The WSM strategy demonstrates improved expert utilization (lower load balancing violation scores) with a slightly higher test language modeling loss. The mean_global_max_violation represents the average of the highest violations across all layers (measuring the severity of "overloaded" experts), while mean_global_min_violation averages the violations for the least-utilized experts (measuring the risk of "routing collapse").

|  | language modeling loss | mean_global_max_violation | mean_global_min_violation |
|---|---|---|---|
| WSD | 0.675 | 0.601 | 0.322 |
| WSM | 0.697 | **0.545** | **0.201** |

We provide an analysis of the implications of WSM on MoE router balance in Table 5. Specifically, the violation for a single expert is calculated as its relative deviation from the average load within its layer[1]. When comparing the merged checkpoint with a decayed checkpoint, the merged checkpoint achieves more balanced routing, although its loss is slightly higher. We argue that this trade-off—a marginally higher loss for superior downstream performance—is indicative of enhanced generalization rather than overfitting to the training data.

## 5 RELATED WORK

### 5.1 LEARNING RATE SCHEDULE

Learning rate (LR) scheduling is critical for training performant models (Jin et al., 2023; Gotmare et al., 2019). Classic schedules like Cosine (Kaplan et al., 2020; Ibrahim et al., 2024) or Linear (Defazio et al., 2023) decay adjust the LR based on a predefined total training duration, which is inflexible for continual training. The Warmup-Stable-Decay (WSD) schedule (Hu et al., 2024) addresses this by introducing a stable LR phase after warmup, decoupling the eventual decay from a fixed training length and offering greater flexibility for long or continuous training runs. More recently, researches have explored "schedule-free" methods to eliminate the decay phase entirely, which maintain a constant LR and instead leverage weight averaging techniques (Defazio et al., 2024; Song et al., 2025; Zhang et al., 2025). Builds upon these schedule-free principles, our work propose to replace WSD's decay phase with a post-hoc checkpoint merging operation instead of specific online averaging strategies. This simplifies the training pipeline and, by formalizing the connection between LR decay and checkpoint merging, allows decay strategy to be theoretically approximated, leading to free offline exploration and enhanced model performance.

### 5.2 MODEL MERGING

Model merging (Izmailov et al., 2018; Wortsman et al., 2022) has emerged as an efficient paradigm for model construction. This approach achieves effective knowledge transfer and performance improvement through parameter-level integration of multiple models. Model merging is primarily utilized in two distinct scenarios: (1) The integration of knowledge and capabilities from multiple independently trained models into a single parameter set, with the objective of preserving maximal performance from each specialized model (Aakanksha et al., 2025; Ramé et al., 2024); and (2) the merging of checkpoints along a single training trajectory. This second category is formally established in the literature as Stochastic Weight Averaging (SWA)(Izmailov et al., 2018). SWA func-

---

[1]For a single layer, let $L$ be the vector of token loads for its experts and $\mu = \text{mean}(L)$, max_violation = $\frac{|\max(L)-\mu|}{\mu}$ and min_violation = $\frac{|\min(L)-\mu|}{\mu}$

tions as a practical realization of Polyak averaging(Polyak & Juditsky, 1992), acting as a smoothing mechanism that reduces the noise inherent in stochastic gradient-based optimization (Sanyal et al., 2023; Liu et al., 2024a; Kaddour, 2022; Li et al., 2023; Sandler et al., 2023; Hägele et al., 2024). While Li et al. (2025) show model merging can achieve performance competitive with decay-based schedule, these techniques have also demonstrated practical utility in industrial-scale LLM development (Grattafiori et al., 2024; DeepSeek-AI et al., 2024; Aakanksha et al., 2025).

Concurrent with our work, the ERNIE 4.5 (ERNIE-Team, 2025) explored the relationship between EMA and LR decay. Our work generalizes this by introducing a formal theoretical framework that maps any LR decay schedule to principled averaging weights, and we empirically demonstrate that these non-EMA weights significantly outperform the sub-optimal EMA baseline.

Our work builds upon this SWA paradigm. We establish a theoretical connection between this operation and LR decay and provide a principled approach to convert various LR decay strategies into a theoretically approximate model averaging implementation.

## 6   Conclusion

In this paper, we have presented WSM, a decay-free LR scheduling approach for LLM pre-training. Our method bridges LR decay and checkpoint merging by establishing the theoretical connection. By eliminating the conventional decay phase, WSM simplifies LR scheduling while reformulating various decay strategies as principled model averaging schemes. Through systematic analysis, we identified the incorporation of high-quality annealing data and merge duration as the most critical factors influencing model performance—outweighing other implementation choices. Extensive experiments have demonstrated WSM's superiority over traditional WSD baselines in LLM pre-training, with consistent improvements across multiple benchmarks, robustness across different optimizers, and sustained benefits during fine-tuning. Consequently, WSM offers a highly versatile approach that can be seamlessly integrated into existing training pipelines without additional complexity.

## Use of Large Language Models

We utilized large language models (e.g., GPT-5 and Gemini-2.5-pro) during the preparation of this manuscript. Their assistance included editing the text to enhance clarity and readability, as well as generating the plotting code for the figures.

### Acknowledgments

This work was partially supported by the National Natural Science Foundation of China No. 92470205 and Beijing Major Science and Technology Project under Contract No. Z251100008425002. This work was also supported by Ant Group Research Fund. Xin Zhao is the corresponding author.

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

## A   EXPERIMENTAL SETTINGS

**Model architecture**   The core architectures of our experimental model are detailed in Table 6. The model is configured with 20 layers and a hidden dimension size of 2048. Except for the first layer, all FFNs layers are replaced with MoE layers. We adopt the GQA attention mechanism (Ainslie et al., 2023) and integrate Rotary Position Embedding (RoPE) (Su et al., 2024), enabling the model to support sequence lengths up to 8K tokens. For parameter initialization, all learnable parameters are randomly initialized using a standard deviation of 0.006. Under this configuration, the model consists of a total of 16.3 billion parameters, of which approximately 1.43 billion are activated for each token during inference.

Table 6: Detailed model architectures.

| $n_{layers}$ | $d_{model}$ | $d_{ffn}$ | $d_{expert}$ | $n_{heads}$ | $n_{kv\_head}$ | $E$ | $E_a$ | $E_s$ | $N$ | $N_a$ |
|---|---|---|---|---|---|---|---|---|---|---|
| 20 | 2048 | 5120 | 512 | 16 | 4 | 256 | 8 | 1 | 16.3B | 1.43B |

**Training hyperparameters**   We use the AdamW optimizer (Loshchilov & Hutter, 2019) with hyperparameters set as follows: $\beta_1 = 0.9$, $\beta_2 = 0.95$, and weight decay of 0.1. Gradient clipping (Zhang et al., 2020) norm is set to 1.0. According to the scaling laws for MoE optimal hyperparameters, the maximum learning rates were set to $3.74e-4$. The batch size is set to 2048, and with a maximum sequence length of 8K, each training batch contains 16M tokens.

**Pre-training dataset**   The training data is sourced from a large-scale multilingual corpus, primarily covering English and Chinese, while also including various other languages. This corpus encompasses web text, mathematical materials, programming scripts, published literature, and diverse textual content. To validate model performance, we extracted a 10T-token subset from this corpus for training.

**Evaluation setup**   To evaluate performance, we consider a diverse suite of downstream tasks designed to provide a holistic assessment of model capabilities. For base model, tasks are grouped into several categories, such as: (a) General Knowledge/Reasoning (e.g., ARC (Bhakthavatsalam et al., 2021), AGIEval (Zhong et al., 2024), OpenBookQA (Mihaylov et al., 2018), BBH (Suzgun et al., 2023), WorldSense (Hong et al., 2025), PIQA (Bisk et al., 2020), hellaswag (Zellers et al., 2019) and KOR-Bench (Ma et al., 2025)) (b) Language Understanding (e.g., race (Lai et al., 2017), SQuAD 2.0 (Rajpurkar et al., 2018), TriviaQA (Joshi et al., 2017), NQ (Kwiatkowski et al., 2019) and winogrande (Sakaguchi et al., 2021)) (c) Professional Knowledge (e.g., MMLU (Hendrycks et al., 2021a), CMMLU (Li et al., 2024a), C-Eval (Huang et al., 2023), MMLU-Pro (Wang et al., 2024b), GPQA (Rein et al., 2023) and SuperGPQA (Team et al., 2025)) (d) Math (e.g., GSM8K (Cobbe et al., 2021), MATH (Hendrycks et al., 2021b), gaokao (Zhang et al., 2023), GSM-Plus (Li et al., 2024b), mgsm-zh (Shi et al., 2023), CMATH (Wei et al., 2023), MathBench (Liu et al., 2024b), minerva_math (Hendrycks et al., 2021b), college_math (Tang et al., 2024) and cn_middle_school_24) (e) Code (e.g., HumanEval (Chen et al., 2021), LiveCodeBench (Jain et al., 2025), MBPP (Tao et al., 2024), HumanEval_plus (Liu et al., 2023), MBPP_plus (Liu et al., 2023), HumanEval_cn (Peng et al., 2024), HumanEval_fim (Bavarian et al., 2022) and CruxEval (Gu et al., 2024)). For SFT model, the categories and tasks are shown in Table 9 and 10.

## B   DETAILS OF THEORETICAL CONNECTION BETWEEN DECAY AND MERGING

The core idea of checkpoint merging in this work is to take an ordered list of checkpoints, $[\theta_n, \theta_{n+1}, \ldots, \theta_{n+k}]$, and apply a merge function to generate a single model $\hat{\theta}_{n+k}$. Here, $\theta_i \in \mathbb{R}^d$ represents the model's parameter vector at the $i$-th training iteration.

The most general form is a weighted average of the checkpoints:

$$\hat{\theta}_{n+k} = \sum_{j=0}^{k} c_j \theta_{n+j} \tag{6}$$

where $\{c_j\}$ are non-negative weights that sum to one, i.e., $\sum_{j=0}^{k} c_j = 1$.

While intuitive, this formulation obscures a deeper connection to the training dynamics. We can reveal this link by expressing each checkpoint in terms of an initial checkpoint $\theta_n$ and the subsequent gradient updates.

We assume that the updates between checkpoints at different time steps are independent. Let $g_i$ be the gradient update vector (including the learning rate) at step $i$, such that the model is updated via $\theta_{i+1} = \theta_i - g_i$. Any checkpoint $\theta_{n+j}$ can therefore be written as the sum of an initial state $\theta_n$ and the sequence of negative gradient updates that followed:

$$\theta_{n+j} = \theta_n - \sum_{l=1}^{j} g_{n+l-1} \tag{7}$$

Substituting Eq. 7 into the general merging formula (Eq. 6) yields:

$$\hat{\theta}_{n+k} = \sum_{j=0}^{k} c_j \left( \theta_n - \sum_{l=1}^{j} g_{n+l-1} \right) \tag{8}$$

$$= \left( \sum_{j=0}^{k} c_j \right) \theta_n - \sum_{j=0}^{k} c_j \sum_{l=1}^{j} g_{n+l-1} \tag{9}$$

$$= \theta_n - \sum_{j=1}^{k} c_j \sum_{l=1}^{j} g_{n+l-1} \tag{10}$$

The double summation in Eq. 10 can be rearranged by changing the order of summation. A gradient update $g_{n+i-1}$ is included in the sum for all checkpoints $\theta_{n+j}$ where $j \geq i$.

$$\sum_{j=1}^{k} c_j \sum_{l=1}^{j} g_{n+l-1} = \sum_{i=1}^{k} \left( \sum_{j=i}^{k} c_j \right) g_{n+i-1} \tag{11}$$

This shows that a weight coefficient of $\sum_{j=i}^{k} c_j$ is applied to the gradient update $g_{n+i-1}$ from step $n+i-1$. By defining a new set of weights for the gradient updates, $w_i = \sum_{j=i}^{k} c_j$, we arrive at the final equivalent expression for checkpoint merging:

$$\hat{\theta}_{n+k} = \theta_n - \sum_{i=1}^{k} w_i \cdot g_{n+i-1} \tag{12}$$

This equation demonstrates that merging checkpoints with weights $\{c_j\}$ is equivalent to applying a synthetic decay schedule defined by weights $\{w_i\}$ to the gradients accumulated after the base checkpoint $\theta_n$, and there exists a mapping between the merging weights $c_j$ and the effective learning rates $w_i$

### B.1 PROOF OF THEOREM 3.1

We seek to find the unique checkpoint weights $\{c_j\}$ corresponding to a given desired sequence of gradient decay coefficients $\{w_i\}_{i=1}^{k}$. The relationship derived in the previous section is the starting point:

$$w_i = \sum_{j=i}^{k} c_j \tag{13}$$

We can solve for the checkpoint weights $\{c_j\}$ by starting from the last element and working backwards.

For $i = k$, the sum in Eq. 13 has only one term:

$$w_k = \sum_{j=k}^{k} c_j = c_k \tag{14}$$

This gives us the value of $c_k$ directly.

For any $i \in [1, k-1]$, we can write out the expressions for $w_i$ and $w_{i+1}$:

$$w_i = c_i + c_{i+1} + c_{i+2} + \cdots + c_k$$
$$w_{i+1} = \phantom{c_i +} c_{i+1} + c_{i+2} + \cdots + c_k$$

Subtracting the second equation from the first yields the expression for $c_i$:

$$w_i - w_{i+1} = c_i \tag{15}$$

Finally, for $c_0$, we use the constraint that the checkpoint weights must sum to one: $\sum_{j=0}^{k} c_j = 1$.

$$
\begin{aligned}
c_0 &= 1 - \sum_{j=1}^{k} c_j \\
&= 1 - (c_1 + c_2 + \cdots + c_{k-1} + c_k) \\
&= 1 - ((w_1 - w_2) + (w_2 - w_3) + \cdots + (w_{k-1} - w_k) + w_k)
\end{aligned}
\tag{16}
$$

The sum in the parentheses is a telescoping series which simplifies to $w_1$.

$$c_0 = 1 - w_1 \tag{17}$$

This completes the derivation of the unique formulas for $\{c_j\}$ as stated in the theorem.

For the checkpoint weights $\{c_j\}$ to be valid, they must be non-negative. This imposes conditions on the sequence $\{w_i\}$.

- From Eq. 14, $c_k \geq 0$ implies $w_k \geq 0$.
- From Eq. 15, $c_j \geq 0$ for $j \in [1, k-1]$ implies $w_j - w_{j+1} \geq 0$, which means $w_j \geq w_{j+1}$. This shows that the sequence $\{w_i\}$ must be monotonically non-increasing.
- From Eq. 17, $c_0 \geq 0$ implies $1 - w_1 \geq 0$, which means $w_1 \leq 1$.

Combining these conditions, we arrive at the requirement that the gradient decay sequence must be bounded and monotonically non-increasing: $1 \geq w_1 \geq w_2 \geq \cdots \geq w_k \geq 0$. This ensures that a valid (non-negative) set of checkpoint weights $\{c_j\}$ can be derived.

## C  DISCUSSIONS

**Choice between Offline and Online Merging**  We prioritize an offline, checkpoint-based approach over standard online running averages (such as maintaining a recursive average at every step) for two key reasons:  **(1) Flexibility:** WSM allows us to explore dozens of distinct virtual decay schedules from a single training run. By preserving checkpoints, practitioners can systematically evaluate how different factors influence the final model, including the choice of merging algorithm (simulating various decay curves like linear or 1-*sqrt*) and the annealing duration (by varying the merge window size $n$). In contrast, an online average commits to a single annealing path. If that trajectory proves suboptimal, the only recourse is expensive retraining. **(2) Performance:** Standard online running averages typically behave as an Exponential Moving Average (EMA). As shown in our comparisons, EMA is a suboptimal merging strategy compared to the concave schemes (e.g., 1-*sqrt*) enabled by WSM. A recursive online average is fundamentally incapable of simulating these more effective non-uniform schedules. However, once this offline exploration identifies a superior, fixed strategy (e.g., "an average of the last 4 checkpoints"), it can be operationalized as a simple and efficient online process using a sliding window.

**Practicality and Complexity**  Compared to discovering an optimal decay schedule through multiple, resource-intensive training runs, the merging operations in the WSM strategy save substantial computational resources. The main complexity introduced is storage overhead. For the offline merging approach, which is ideal for exploration, one must store a history of checkpoints, with the total number being the total training tokens divided by the checkpointing interval. However, this is a

manageable trade-off, as this storage footprint represents a minor fraction of a typical pre-training budget. Furthermore, once a superior strategy is identified, or in scenarios with extreme storage constraints, an online merging approach can be used. This method minimizes the storage footprint by maintaining only a small, fixed-size window of $n$ checkpoints. Our experiments confirm that a relatively small window (e.g., $n = 12$) is sufficient to achieve strong results.

**Comparison with Concurrent Work.** We acknowledge concurrent work that has explored model merging during pre-training, most notably from the EMA-based approach in ERNIE 4.5 (ERNIE-Team, 2025) and PMA (Li et al., 2025). We highlight the key differences between our work and theirs as follows: (1) **Motivation**: We frame checkpoint merging as a novel LR scheduling mechanism, with the primary goal of discovering a schedule that outperforms strong baselines like WSD. In contrast, their work treats model merging as a standalone pre-training technique, evaluating its value by comparing the performance of the merged model against its before-merge checkpoints. (2) **Methodology**: We propose a general theoretical framework that generalizes beyond simple EMA. While their discussion centers almost exclusively on EMA, our framework formalizes a mapping from any LR decay schedule (e.g., linear, inverse-sqrt, cosine) to a specific, principled set of averaging weights. (3) **Key Findings**: Our empirical results demonstrate that merge weights corresponding to linear and 1-sqrt decay schedules yield significantly better performance, while the EMA baseline is sub-optimal. Additionally, we identify the merge duration as the most critical hyperparameter influencing the final model. Ultimately, our method surpasses the performance of the WSD baseline. In contrast, their work concludes that checkpoint merging can effectively match the performance of an optimally-annealed model. We view the findings from our work and these concurrent studies as complementary, and together they offer valuable practical insights for LLM pre-training.

**Distinction from Prior Theoretical Works** While previous studies (Bergsma et al., 2025; Wang et al., 2024a) have noted theoretical connections between learning rate schedules and weight averaging, our work differs fundamentally in both objective and scope. **(1) Paradigm Shift vs. Analysis:** Prior works primarily utilize this connection as an analytical tool to explain the convergence properties of standard decay schedules. Their goal is to *analyze* the existing paradigm. In contrast, our objective is to *replace* it. We propose WSM as a standalone training strategy that eliminates the need for an explicit decay phase. We are, to our knowledge, the first to systematically validate offline averaging as a direct, efficient substitute for LR decay in large-scale pre-training. **(2) General Framework vs. EMA:** Previous theoretical discussions have focused almost exclusively on the relationship between SGD and Exponential Moving Average (EMA). Our framework generalizes this significantly by providing a constructive proof that maps *any* decay schedule (e.g., linear, 1-*sqrt*) to a specific set of principled averaging weights. This distinction is practically vital: our experiments demonstrate that EMA is often a sub-optimal strategy. The superior performance of WSM relies directly on the precise, non-EMA weights derived from our general framework, allowing us to simulate concave schedules that standard EMA cannot approximate.

## D    ADDITIONAL EXPERIMENTS ON MUON OPTIMIZER

To empirically validate the applicability of WSM across different optimization algorithms, we conducted an additional experiment using the Muon optimizer (Jordan, 2024), a distinct representative optimizer compared to the AdamW optimizer used in our main experiments. We performed a 100B token enhancement training session initialized from checkpoints trained on 2T tokens. The evaluation metrics and benchmarks remain consistent with those described in the main paper. The comparative results between WSD and WSM are presented in Table 7. We observe that WSM achieves significant gains over WSD across the majority of domains. These findings demonstrate that the effectiveness of WSM is not limited to specific optimizers and maintains its superiority in diverse optimization landscapes.

Table 7: Performance comparison with Muon optimizer on 100B-token enhancement training (initialized from 2T checkpoints).

|  | General Knowledge/ Reasoning | Language Understanding | Math | Code | Professional Knowledge | Overall Average |
|---|---|---|---|---|---|---|
| WSD | 63.37 | 75.50 | 55.01 | 47.10 | **45.09** | 56.10 |
| WSM | **63.95** | **76.95** | **56.85** | **47.30** | 45.04 | **56.74** |

# E  ALGORITHM

---

**Algorithm 1** The WSM LRS Pre-training Pipeline

---

**Input:** Initial model weights $W_0$, pre-training data $D$, high-quality annealing data $D_{anneal}$ (optional), peak LR $lr_{peak}$, warmup steps $T_{warmup}$, switch step $T_{switch}$, checkpointing interval $T_{cpt}$, merging window size $n$

**Initialize:**
▷ The merged checkpoint.
$W_{merged} \leftarrow$ null
▷ Storage for the base (un-merged) checkpoints.
$\mathcal{C}_{storage} \leftarrow []$

▷ **Phase 1: Warmup on Pre-training Data**
**for** $t = 1$ to $T_{warmup}$ **do**
    $lr(t) \leftarrow lr_{peak} \cdot (t/T_{warmup})$
    $W_t \leftarrow \text{Update}(W_{t-1}, D, lr(t))$
**end for**

▷ **Phase 2: Stable Training and Merging**
**for** $t = T_{warmup} + 1$ to ... **do**
    $lr(t) \leftarrow lr_{peak}$
    ▷ Select dataset for the current step
    **if** $t > T_{switch}$ **and** $D_{anneal}$ is available **then**
        $D_{current} \leftarrow D_{anneal}$
    **else**
        $D_{current} \leftarrow D$
    **end if**
    $W_t \leftarrow \text{Update}(W_{t-1}, D_{current}, lr(t))$
    **if** $t \pmod{T_{cpt}} == 0$ **then**
        ▷ Save the checkpoint from the main training process.
        $\text{SaveToCheckpointStorage}(\mathcal{C}_{storage}, W_t)$
        ▷ **Asynchronously Update the Merged checkpoint**
        $C_{latest} \leftarrow \text{GetLastNCheckpoints}(\mathcal{C}_{storage}, n)$
        **if** $\text{len}(C_{latest}) == n$ **then**
            ▷ Update the merged checkpoint for evaluation.
            $W_{merged} \leftarrow \text{Merge}(C_{latest})$
        **end if**
    **end if**
**end for**
**return** The stored base checkpoints in $\mathcal{C}_{storage}$ and the merged checkpoint $W_{merged}$.

---

# F  ADDITIONAL EXPERIMENTS ON WSD

Our preliminary experiments into Warmup-Stable-Decay (WSD) learning rate schedules revealed a clear performance hierarchy among different decay curves, with the 1-sqrt strategy emerging as superior. Specifically, we conducted a controlled experiment initialized from a base checkpoint that was pre-trained on 7T tokens. We then annealed the model for an additional 100B tokens using a consistent, high-quality dataset, where we varied only the annealing decay function. The final learning rate for all experimental runs was set to zero. The results, depicted in Figure 6, confirm that

the 1-*sqrt* decay outperforms other methods in benchmarks. Based on this evidence, we establish the WSD schedule with 1-sqrt decay as a strong baseline for all subsequent experiments. As this was a preliminary study, the starting checkpoint used here differs from that used in our main experiments in Section 4.

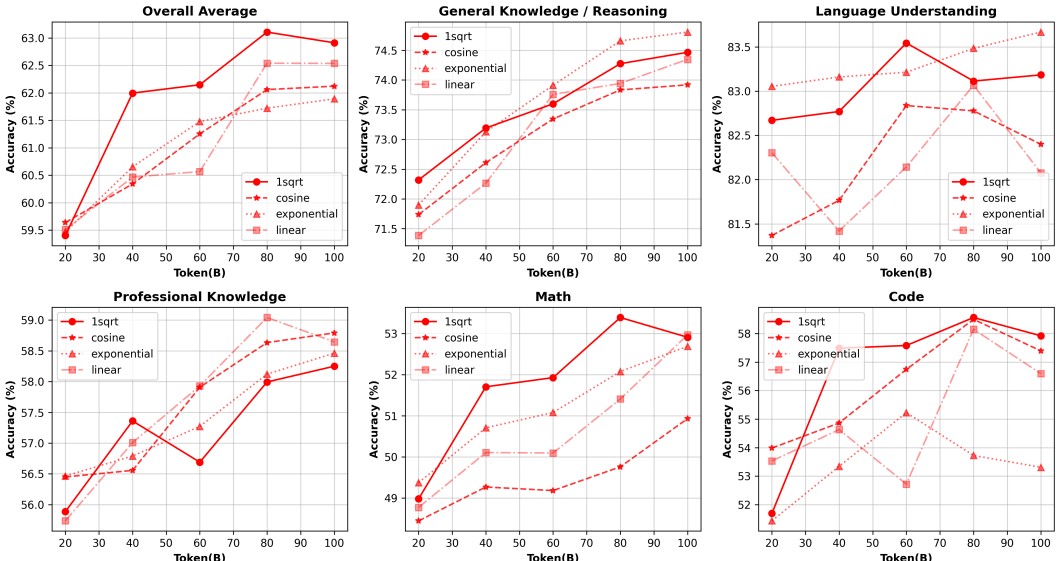

Figure 6: Comprehensive performance comparison between different decay strategy of WSM schedule.

# G    HOW CHECKPOINT MERGING INFLUENCES NETWORK BEHAVIOR: STABILITY & GENERALIZABILITY

The architecture composed of sequentially connected transformer blocks has been widely adopted in mainstream LLMs. In such chain-like structures, variations in shallow-layer outputs typically propagate layer-wise. More stable and generalizable input-output patterns exhibit greater potential for enhancing downstream task performance. To investigate how checkpoint merging influences network behavior, we conduct the following analyses:

**Stability**    In numerical analysis, the condition number quantifies a function's sensitivity to input perturbations and the resultant output errors. Remarkably, our analysis in Figure 7 reveals that the decay process induces sharp deterioration of condition numbers, whereas checkpoint merging demonstrates superior stability. This indicates that the checkpoint merging strategy not only improves performance but also preserves parametric stability.

**Generalizability**    When maintaining competitive performance, higher SVD entropy (singular values entropy) correlates with greater matrix effective rank, indicating greater information capacity in matrix operations. For continue pre-training and fine-tuning scenarios, higher SVD Entropy often means higher potential for model plasticity (Alter et al., 2000; Roy & Vetterli, 2007; Liu et al., 2025). Figure 8 shows the trend of SVD Entropy during training. We observe that decay is a rapid entropy reduction process, which continuously damages the potential for future continue training of the model. In contrast, the merged models still maintain a higher generalizability, manifested as a higher SVD Entropy.

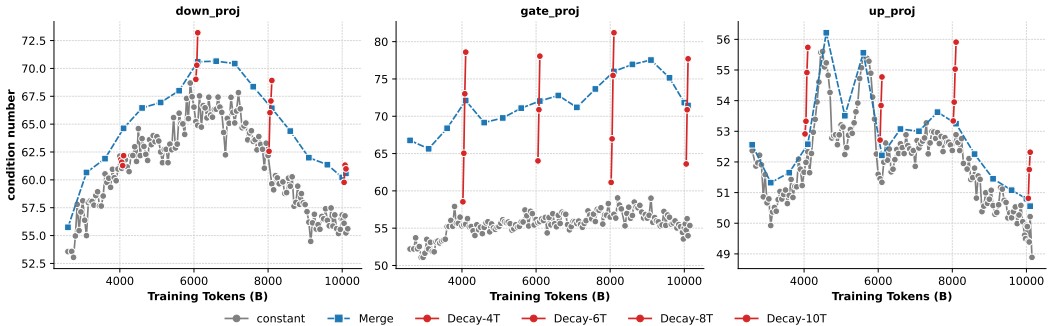

Figure 7: Condition number of weight matrices across different training tokens.

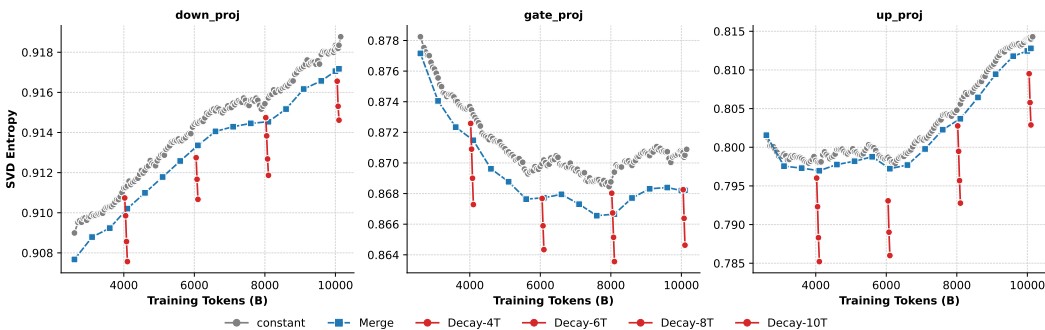

Figure 8: The entropy of singular values for each weight matrix in the feed-forward network (FFN) layers.

## H  PARAMETER TRAJECTORIES AND MODEL PERFORMANCE: CONSTANT LR WITH MERGE VS. LR DECAY

To visually analyze the correlation between model parameters and performance during training, we employ t-SNE (van der Maaten & Hinton, 2008) dimensionality reduction to project the weight matrix of a specific layer into the 2-dimensional embedding space. This projection is combined with performance evaluations to generate the composite contour map illustrated in Figure 9. The directional arrows in the visualization explicitly illustrate the parameter trajectories across both constant and decay phases. Our experimental analysis revealed two principal findings: 1) During the decay phase, model parameters gradually converge toward the merged model solution space, which achieves superior performance compared to checkpoints with a constant LR. 2) The LR reduction in the decay phase enables more precise parameter refinement than the expansive exploration observed under constant LR. This controlled convergence facilitates localization of nearby optimal solutions. We confirm the speculation of previous work Wen et al. (2024), and formalize this dynamic with an analogy: constant LR training resembles traversing a "canyon" with oscillating steps, while merging resembles finding a "river" at the canyon floor that guides efficient convergence.

## I  COMPARISON WITH A LONG-COOLDOWN WSD BASELINE

To ensure a rigorous comparison and align with best practices for WSD schedules, which often recommend cooldown durations of at least 10% of total training tokens, we conducted a controlled experiment to evaluate WSM against a well-tuned WSD baseline with a significantly longer decay period. For this experiment, we started from a 500B-token pre-trained checkpoint and continued training for an additional 200B tokens. For the WSD baseline, the learning rate was exponentially decayed to 10% over this entire 200B-token phase, resulting in a cooldown duration of approximately 28.6% (200B / 700B total tokens). This comfortably satisfies the conventional "¿10% steps" recommendation. For the WSM baseline, we applied our proposed merging strategy to checkpoints

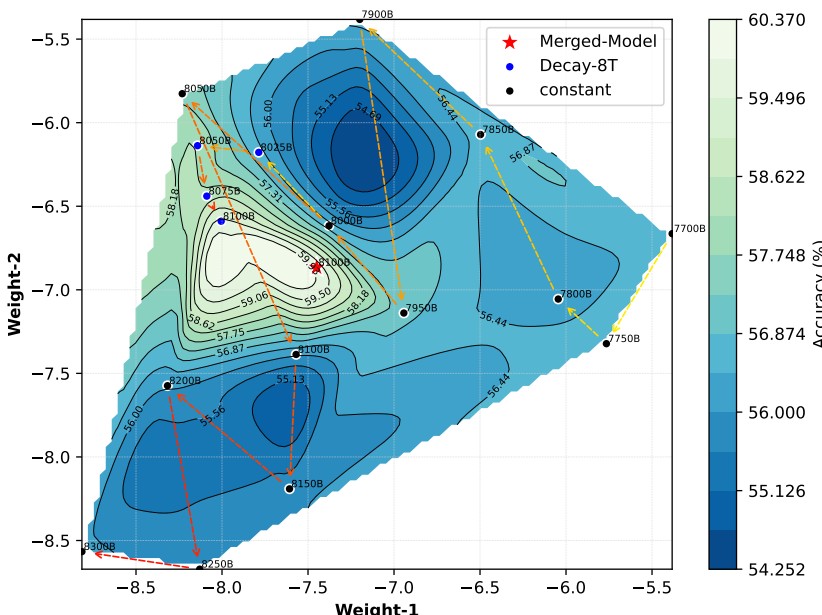

Figure 9: Visualization of Model Parameters and Performance Contour Lines. Points connected by directional arrows represent parameter trajectories during the constant and decay phases, respectively. The red star indicates the solution space of the merged model.

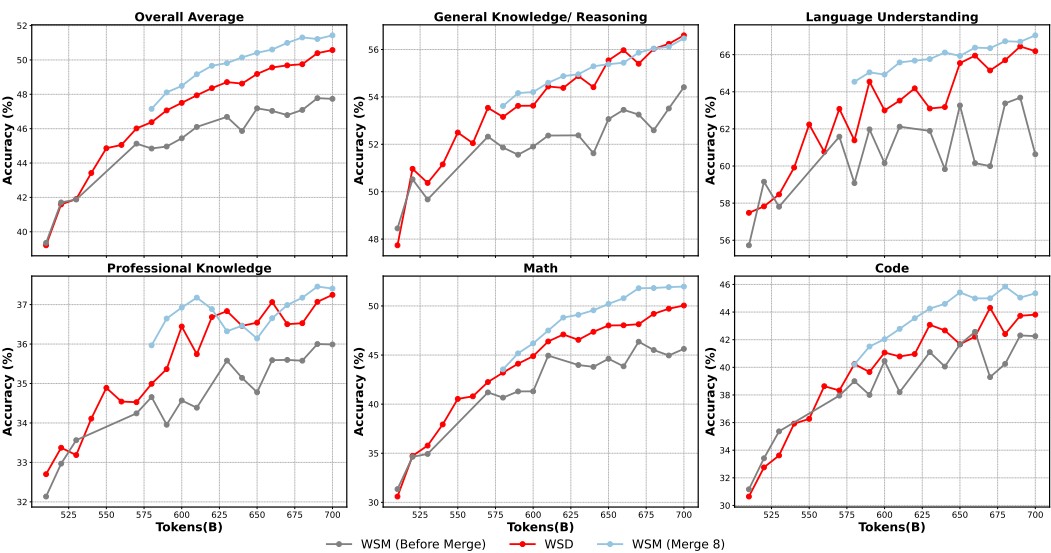

Figure 10: Comprehensive performance comparison between our WSM schedule and standard WSD scheduling on larger decay duration. Starting from a 500B-token checkpoint, we continued training for an additional 200B tokens, resulting in a cooldown ratio of 28.6% (200B / 700B total),

saved during the same 200B-token training phase. The performance of both methods was evaluated on our suite of downstream benchmarks, with the results presented in Table 4. As shown, WSM achieves a higher Overall Average score (51.43 vs. 50.57) and outperforms the WSD baseline in four out of the five sub-categories: Language Understanding, Professional Knowledge, Math, and Code. While WSD shows a marginal advantage in General Knowledge/Reasoning, the overall effectiveness of WSM remains superior. Figure 10 illustrates the training dynamics.

## J   DETAILED EVALUATION RESULTS

We provide detailed evaluation results to compare our methods with WSD. Figure 11 shows a detailed comparison on each dataset across the various categories from the main experiments in Section I, where WSM achieves an advantage on the vast majority of datasets. We select the checkpoint with the highest average score for each method, including WSD and the three WSM merging algorithms, and list them in Table 8. Table 9 and 10 shows the performance comparison of checkpoints generated by the WSM and WSD schedules after supervised fine-tuning (SFT).

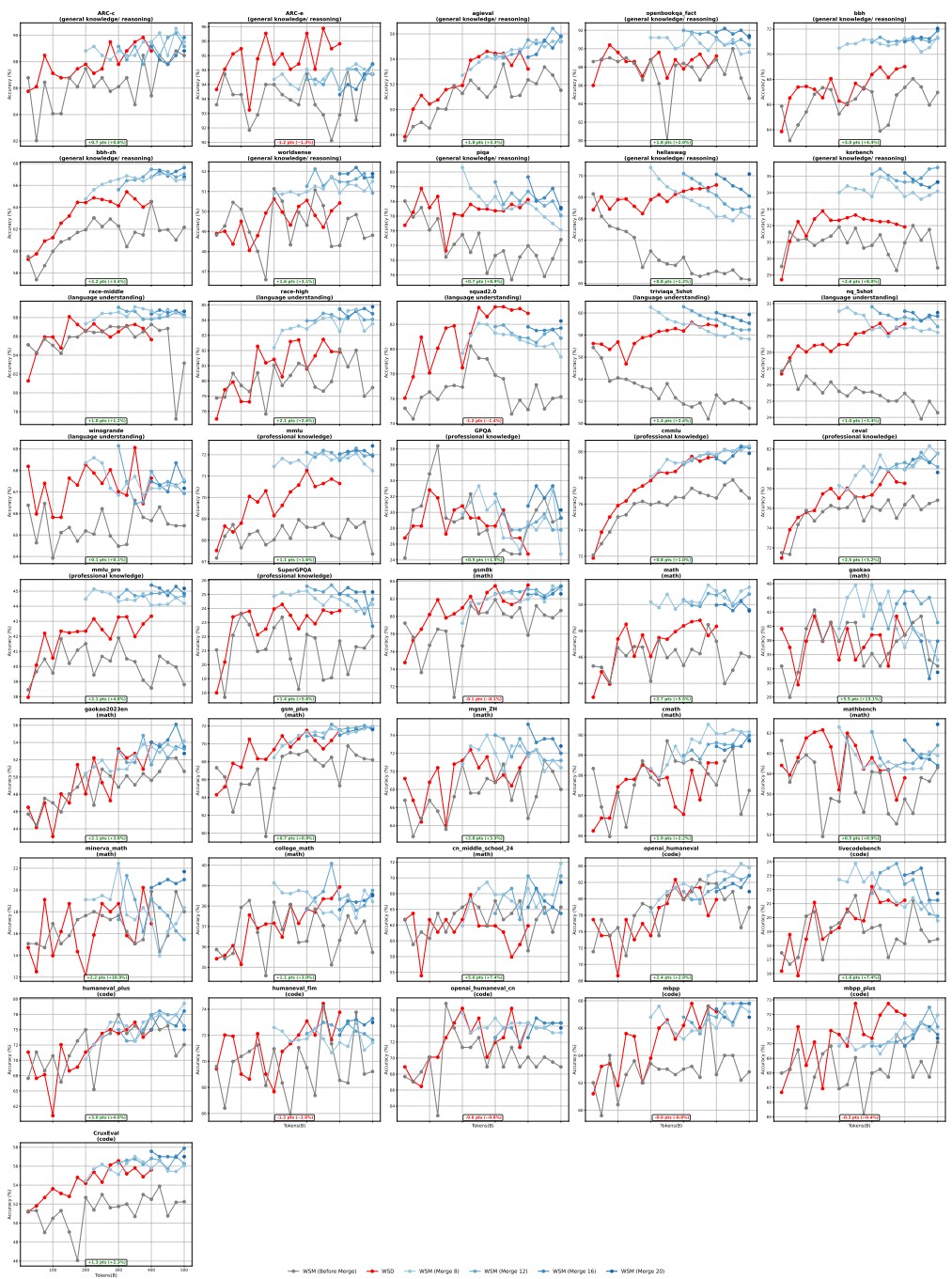

Figure 11: Detailed performance comparison between the standard WSD schedule (via learning rate decay) and our WSM schedule (via checkpoint merging) under different merging durations.

Table 8: Detailed performance comparison of base models trained using WSM (with three distinct merging algorithms) versus WSD scheduling approaches.

| | Metric | WSD | WSM | | |
| --- | --- | --- | --- | --- | --- |
| | | | EMA | mean | 1-*sqrt* |
| General Knowledge & Reasoning | ARC-c | 88.8 | 87.8 | 88.1 | 89.8 |
| | ARC-e | 95.4 | 93.7 | 94.0 | 94.5 |
| | AGIEval | 53.2 | 53.8 | 54.4 | 54.2 |
| | OpenBookQA_fact | 89.2 | 89.2 | 92.2 | 91.8 |
| | BBH | 69.0 | 70.0 | 71.2 | 71.0 |
| | BBH-zh | 64.5 | 65.3 | 67.3 | 66.5 |
| | WorldSense | 50.4 | 50.6 | 51.9 | 52.1 |
| | PIQA | 78.6 | 78.2 | 78.5 | 78.5 |
| | HellaSwag | 69.6 | 68.2 | 69.9 | 69.7 |
| | KOR-Bench | 31.9 | 33.7 | 34.8 | 34.6 |
| Language Understanding | race-middle | 85.7 | 88.7 | 88.8 | 87.9 |
| | race-high | 81.9 | 82.2 | 84.2 | 83.9 |
| | SQuAD2.0 | 82.9 | 81.6 | 81.5 | 81.3 |
| | TriviaQA_5shot | 58.8 | 57.5 | 59.8 | 59.7 |
| | NQ_5shot | 29.8 | 29.1 | 30.4 | 30.3 |
| | WinoGrande | 67.6 | 66.7 | 67.3 | 66.6 |
| Professional Knowledge | MMLU | 70.7 | 71.2 | 72.2 | 72.1 |
| | GPQA | 24.8 | 30.8 | 33.3 | 30.8 |
| | CMMLU | 79.6 | 78.3 | 79.2 | 79.3 |
| | MMLU-Pro | 43.3 | 43.8 | 45.2 | 44.7 |
| | SuperGPQA | 23.8 | 24.3 | 25.5 | 25.7 |
| | C-Eval | 78.5 | 78.3 | 80.9 | 79.9 |
| Math | gsm8k | 83.6 | 82.0 | 82.6 | 82.3 |
| | MATH | 48.3 | 50.8 | 50.0 | 49.6 |
| | gaokao | 38.5 | 47.3 | 39.6 | 48.4 |
| | gaokao2023en | 54.0 | 49.1 | 53.5 | 53.0 |
| | gsm_plus | 71.5 | 69.3 | 71.4 | 70.9 |
| | mgsm_zh | 72.0 | 73.2 | 73.2 | 72.8 |
| | CMATH | 88.6 | 88.4 | 89.3 | 89.2 |
| | MathBench | 57.6 | 61.8 | 60.3 | 61.6 |
| | minerva_math | 16.9 | 19.9 | 20.6 | 19.1 |
| | college_math | 38.9 | 37.8 | 38.3 | 39.0 |
| | cn_middle_school_24 | 62.4 | 67.3 | 68.3 | 70.3 |
| Code | HumanEval | 79.9 | 80.5 | 81.7 | 81.1 |
| | LiveCodeBench | 21.2 | 20.3 | 23.2 | 22.7 |
| | HumanEval_plus | 75.0 | 75.6 | 77.4 | 77.4 |
| | HumanEval_fim | 73.8 | 69.3 | 73.1 | 73.9 |
| | HumanEval_cn | 74.4 | 77.4 | 75.0 | 75.6 |
| | MBPP | 67.2 | 65.0 | 66.8 | 66.8 |
| | MBPP_plus | 72.0 | 70.6 | 70.4 | 70.6 |
| | CruxEval | 55.6 | 54.8 | 57.0 | 57.4 |

Table 9: Detailed performance comparison of checkpoints generated by the WSM and WSD schedule after supervised fine-tuning (SFT). Both base checkpoints are fine-tuned under identical settings for 5 epochs. Results are reported based on the epoch with the highest average benchmark score.

| | | Metric | WSD | WSM |
|---|---|---|---|---|
| Knowledge | Basic Knowledge | ARC-c | 90.85 | 89.49 |
| | | BoolQ | 84.80 | 85.38 |
| | | GaokaoBench | 75.70 | 79.95 |
| | | AGIEval | 61.87 | 65.22 |
| | | NQ | 25.4 | 26.43 |
| | | TriviaQA | 53.93 | 55.52 |
| | Average | | 65.42 | 67.00 |
| | Professional Knowledge | C-Eval | 76.37 | 77.87 |
| | | CMMLU | 76.13 | 76.78 |
| | | MMLU | 72.76 | 74.59 |
| | | MMLU-Pro | 46.09 | 49.67 |
| | | GPQA | 29.55 | 33.4 |
| | | SuperGQPA | 26.50 | 26.43 |
| | Average | | 54.57 | 56.46 |
| Code | Code Completion | HumanEval | 85.90 | 86.20 |
| | | HumanEval_plus | 81.10 | 80.95 |
| | | HumanEval_cn | 78.20 | 76.07 |
| | | CruxEval | 59.69 | 58.88 |
| | | Multiple | 66.14 | 64.75 |
| | | HumanEvalFix | 63.01 | 62.50 |
| | Average | | 72.34 | 71.56 |
| | Code Generation | MBPP | 81.03 | 81.97 |
| | | MBPP_plus | 73.02 | 71.43 |
| | | LiveCodeBench | 31.31 | 33.31 |
| | | BigCodeBench | 33.77 | 33.16 |
| | | CodeForces | 19.60 | 18.07 |
| | | Bird-SQL | 25.95 | 28.10 |
| | Average | | 44.11 | 44.34 |
| Math | Elementary Mathematics | CMATH | 93.08 | 93.99 |
| | | gsm8k | 87.34 | 87.79 |
| | | cn_middle_school_24 | 72.28 | 73.27 |
| | | mgsm_zh | 76.80 | 78.25 |
| | | gsm_plus | 77.28 | 77.93 |
| | Average | | 81.36 | 82.25 |
| | Intermediate Mathematics | MATH | 74.24 | 77.08 |
| | | MathBench | 74.72 | 75.74 |
| | | college math | 43.57 | 43.52 |
| | | gaokao | 64.17 | 60.77 |
| | | minerva math | 55.74 | 57.31 |
| | | gaokao2023en | 63.12 | 65.19 |
| | | MATH500 | 74.75 | 77.05 |
| | Average | | 64.33 | 65.24 |
| | Advanced Mathematics | OlympiadBench | 41.89 | 41.93 |
| | | AIME2024 | 11.04 | 11.67 |
| | | AIME2025 | 11.46 | 12.71 |
| | Average | | 21.46 | 22.10 |

Table 10: Detailed performance comparison of checkpoints generated by the WSM and WSD schedule after supervised fine-tuning (SFT). Both base checkpoints are fine-tuned under identical settings for 5 epochs. Results are reported based on the epoch with the highest average benchmark score.

| | | **Metric** | **WSD** | **WSM** |
|---|---|---|---|---|
| Language | Language Understanding | C3 | 84.38 | 87.18 |
| | | WSC | 69.23 | 77.88 |
| | | race-high | 82.93 | 84.65 |
| | | race-middle | 87.95 | 89.42 |
| | Average | | 81.12 | 84.78 |
| Reasoning | Complex Reasoning | bbh | 72.87 | 73.97 |
| | | drop | 76.41 | 78.66 |
| | | hellaswag | 68.11 | 70.93 |
| | | ocnli | 51.32 | 52.75 |
| | | piqa | 81.50 | 82.92 |
| | | ProntoQA | 37.00 | 38.00 |
| | | Multi-LogiEval | 0.27 | 56.68 |
| | | MuSR | 48.53 | 51.33 |
| | | korbench | 37.52 | 37.52 |
| | | bbh-zh | 69.38 | 71.47 |
| | Average | | 63.21 | 64.94 |
| Agent | Tool-use | teval_v2_en | 84.4 | 86.16 |
| | | teval_v2_zh | 83.49 | 84.72 |
| | | BFCL_AST | 79.35 | 78.30 |
| | | BFCL-Live | 68.33 | 71.39 |
| | | NEXUS | 30.51 | 29.21 |
| | Average | | 69.22 | 69.96 |
| | Instruction Following | IFEval | 71.9 | 75.71 |
| | | alignbench | 59.10 | 59.80 |
| | Average | | 65.5 | 67.75 |

