# OpenReview forum: "WSM: Decay-Free Learning Rate Schedule via Checkpoint Merging for LLM Pre-training"
_ICLR.cc/2026/Conference — ICLR 2026 Oral_

### Official Review · Reviewer_852L · 2025-10-15

**Soundness:** 3
**Presentation:** 3
**Contribution:** 3
**Rating:** 6
**Confidence:** 4

**Summary:**

The paper introduces Warmup-Stable and Merge (WSM), a novel and compelling learning rate (LR) scheduling framework for pre-training large language models. The core idea is to replace the conventional, inflexible LR decay phase with a "decay-free" approach. In WSM, after a standard warmup, the model is trained with a constant peak LR (the "stable" phase), during which checkpoints are periodically saved. The final performant model is then produced by applying a weighted average (merging) to a series of these saved checkpoints.
The authors make a significant theoretical contribution by establishing a formal connection between checkpoint merging and LR decay. They demonstrate that a weighted average of checkpoints is mathematically equivalent to applying a synthetic, effective decay schedule to the gradient updates accumulated since the first checkpoint in the merge window. This framework allows for the emulation of various decay curves (e.g., linear, cosine, 1-sqrt) by simply choosing the appropriate checkpoint weights, without ever modifying the live learning rate during training.
Through extensive experiments on a 16.3B parameter MoE model, the paper shows that WSM consistently and significantly outperforms the strong Warmup-Stable-Decay (WSD) baseline on a wide range of benchmarks, including MATH, HumanEval, and MMLU-Pro. The key empirical finding is that the "merge duration" (i.e., the time window over which checkpoints are aggregated) is the most critical factor for performance. The benefits of WSM are shown to persist even after supervised fine-tuning.

**Strengths:**

- This paper is the formal connection it establishes between LR decay and checkpoint merging (Section 3.1), which is insightful.

- The experiments are conducted at a scale that is highly relevant to the LLM community (a 16.3B parameter model trained on trillions of tokens).

- The paper provides a thorough empirical analysis of the WSM schedule. The authors methodically investigate the impact of the merging algorithm (Table 3), merge duration (Figure 4), and merge granularity (Table 4).

**Weaknesses:**

- **Significant Computational and Storage Overhead:**
  - While the paper suggests transitioning to an online, sliding-window approach after identifying an optimal strategy, the initial exploration phase itself represents a significant, non-trivial cost. Can auther give a quantified results？
  - The merging process also incurs computational overhead, specifically in terms of I/O for loading multiple large checkpoints and memory consumption, which is not fully quantified in the paper.

- **Insufficient Baselines for Model Averaging:**
  - While the paper compares its theoretically-derived merging weights against mean averaging and EMA, it overlooks other well-established trajectory averaging methods. A direct comparison with classic techniques like Stochastic Weight Averaging (SWA) would be highly valuable.

- **Shift in Hyperparameter Complexity:**
  - The paper claims that WSM simplifies LR scheduling by eliminating the decay phase. However, it arguably replaces one set of hyperparameters (decay function, duration) with another, potentially more complex set: merge duration/window size, checkpoint interval, and merge algorithm.
  - The finding that "merge duration" is the most critical factor highlights that WSM does not eliminate the need for extensive tuning; it merely shifts the tuning process from an online LR schedule design to an offline search for the optimal merging strategy. This trade-off should be more explicitly discussed.

**Questions:**

- see weakness

---

> ### Author Response · Authors · 2025-11-20
>
> > **Weakness 1: Significant Computational and Storage Overhead.**
>
> **Response:**  We sincerely thank the reviewer for the suggestion. WSM introduces minimal to no additional overhead into a standard large-scale pre-training workflow.
>
> * **Storage Overhead:** In large-scale model training, periodically saving checkpoints is already a mandatory, standard practice for fault recovery, monitoring, and evaluation. WSM simply reuses these existing checkpoints. Therefore, no extra storage is required beyond what is already allocated for the training workflow.
> * **Computational Overhead:** The merging step itself is exceptionally cheap. It involves simple I/O and vectorized arithmetic on the CPU, requires no GPU resources, and its cost is orders of magnitude less than continuing the live training run for even a short period.
>
> Therefore, WSM leverages existing, mature training infrastructure to provide its benefits at virtually no extra cost.
>
> &nbsp;
>
> > **Weakness 2: Insufficient Baselines for Model Averaging.**
>
> **Response:** The core goal of our work is **not** to invent a new trajectory-averaging algorithm, but to demonstrate that many standard LR decay schedules correspond to specific weight-averaging schemes, which can be simulated offline.
> Within this framework, Stochastic Weight Averaging (SWA) is **essentially a special case corresponding to uniform weighting**. Our experiments already include "mean averaging," which is functionally equivalent to SWA over a contiguous window of checkpoints. We will clarify that this corresponds in the revised manuscript.
>
> &nbsp;
>
> > **Weakness 3: Shift in Hyperparameter Complexity.**
>
> **Response:** We argue that WSM does not merely shift complexity but fundamentally transforms it from a **high-cost, high-risk training problem** into a **low-cost, flexible offline optimization** task.
>
> * **Cost Reduction:** In a traditional WSD framework, evaluating each new decay schedule requires launching an entirely new, extremely expensive training run. In contrast, tuning WSM's hyperparameters (e.g., merge duration, schedule type) is done entirely offline, and its total cost is less than 1% of a single training run.
> * **Risk Mitigation:** The most critical advantage of WSM is that it mitigates the irreversible and costly decision of choosing a decay schedule at the start of training. WSM allows this decision to be made after the main training is complete, using real checkpoint data to find the optimal strategy without risk. In an industrial setting where a single training run represents a massive investment, this is a fundamental and decisive advantage.
>
> In short, WSM replaces the need for multiple, costly retraining runs with a single, low-cost offline tuning phase. Exploring different decay schedules becomes a matter of cheap CPU and storage, not expensive GPU re-computation. For industrial applications where a training run represents a massive investment, this ability to **find the optimal configuration risk-free is a decisive advantage.**

---

### Official Review · Reviewer_fGtp · 2025-11-01

**Soundness:** 2
**Presentation:** 3
**Contribution:** 2
**Rating:** 2
**Confidence:** 4

**Summary:**

This paper proposes merging checkpoints as an alternative to learning rate annealing.
In particular, warmup-stable-merge (WSM) is proposed as a replacement for the warmup-stable-decay (WSD) learning rate schedule,
which is common for LLM pre-training.
There is some theory to provide intuition. Results are good.
Some analysis experiments investigate which factors are more important.

**Strengths:**

* WSM is a simple method which provides sufficient improvement over WSD and is convenient enough to implement and use that I support the paper's proposal to adopt it instead of WSD.

* The experiments are comprehensive.

**Weaknesses:**

* This work has limited novelty.
The connection between LR schedules and weight averaging is well-known and has more complete treatment elsewhere [1,2].
This includes almost the entirety of Section 3 and forms the main contribution of the paper.
Further, the theoretical treatment in this paper is rather hand-wavy.

* The proposed methodology is also not surprising, although maybe it hasn't appeared in this packaging before.
Weight averaging is a common practice, often applied with annealing [3].

* The authors identify the "merge duration" as the most significant factor and suggest an offline computation to compute the merged model. However, from the experiments it is not clear why or if merging along the entire training trajectory would hurt. That would be much easier to implement online without memory overhead by maintaining a running average. For example, this is the recommendation in [4].

[1] Straight to zero: Why linearly decaying the learning rate to zero works best for LLMs. ICLR 2025.

[2] How to set AdamW's weight decay as you scale model and dataset size. ICML 2025.

[3] The Llama 3 Herd of Models. See Section 3.4.3.

[4] Post-Hoc Reversal: Are We Selecting Models Prematurely? NeurIPS 2024. Search for SWA.

**Questions:**

Some concerns / questions / comments:

* Section 5.2: What you are calling model merging in this paper is more commonly termed as Stochastic Weight Averaging (SWA). To my knowledge, "model merging" is used in the context where the models come from different training/fine-tuning runs. i.e. (1) in L460 is "model merging" but (2) in L463 is SWA. At the very least, SWA should be mentioned and discussed in the paper as this is the official name used by the 2018 paper (which you have cited under model merging in L458).

* WSD is not exactly schedule-free as it has been found that optimal LR depends on the decay start time [1]. By extension WSM is also not schedule-free. Please fix this claim, or at least add a caveat and discussion.

* L115: You still need / will benefit from LR tuning, so please remove this line.

* L257: To help the readers, please clarify that this corresponds to a linear decay LR schedule.

* What is the intuition for why WSM is better than WSD? Can you do some empirical analysis to support the intuition?

* Why is there a systematic dip in all plots towards the end in Figure 4?

* L348: "this finding reinforces the theoretical connection"... I believe you need Figure 5a style curves to rigorously establish this.

* L429: Is "language modeling loss" column the train loss or test loss?

* L483: The theory is not exactly optimizer-agnostic with a more careful analysis as in [2]. For example, the weight decay constant $\lambda$ is an important part of the SWA weight terms.

[1] Straight to zero: Why linearly decaying the learning rate to zero works best for LLMs. ICLR 2025.

[2] How to set AdamW's weight decay as you scale model and dataset size. ICML 2025.

---

> ### Author Response · Authors · 2025-11-20
> **Rebuttal by Authors (part-1/3)**
>
> >**Weakness 1: This work has limited novelty.**
>
> **Response:**  We respectfully disagree with the assessment of our work's novelty. While learning rate (LR) scheduling is indeed a well-established technique, existing methods are constrained by their reliance on pre-defined decay schedules. Our work breaks from this convention by proposing **a paradigm shift: a decay-free optimization framework that uses model merging to dynamically emulate the effect of LR decay, rather than explicitly prescribing it**. This principled method addresses a gap in the literature of decay-based optimization and represents the central novelty of our paper. We elaborate on this point in two specific aspects:
>
> **1. Fundamental Difference in Goal: Replacing vs. Supplementing LR Decay.**
> - **Our Goal is Paradigm Shift:** We propose offline checkpoint averaging as a principled substitute for the entire LR decay phase, aiming to replace the conventional decay schedule with a more efficient and flexible training paradigm. Our objective is to enable more agile pre-training by eliminating the decay stage.
> - **Prior Work Aims for Analysis:** In contrast, prior works [1,2] leverage related interpretations (e.g., EMA) primarily as an analytical tool to explain why existing techniques like linear decay work or to derive hyperparameter scaling laws. Their goal is to supplement and understand the conventional paradigm of LR decay, not to replace it.
>
> To our knowledge, we are the first to propose and systematically validate using offline averaging as a direct replacement for the LR decay stage in large-scale pre-training.
>
> **2. Novel Theoretical Contribution: A General, Constructive Framework Beyond Simple EMA.**
> - **Prior discussions focus almost exclusively on EMA as a qualitative analogy.** In contrast, we provide a constructive proof that establishes a formal mapping from any decay schedule (linear, 1-sqrt, cosine, etc.) to a specific set of principled averaging weights.
> - **Our experiments demonstrate the practical value of this generalization**, where the widely discussed EMA is a sub-optimal baseline. The superior performance of our method is a direct result of the precise, non-EMA weights derived from our theory. This demonstrates that our theoretical framework is not only novel but also essential for the success of this new paradigm.
>
> We will revise Discussion Section of our manuscript to more clearly delineate our novel contributions from the background. Given these fundamental distinctions, we respectfully ask the reviewer to reconsider the novelty and contribution of our work.
>
> &nbsp;
>
> >**Weakness 2: The proposed methodology is also not surprising.**
>
> **Response:** We respectfully disagree with the premise that a method must be "surprising" to be valuable. **The primary goal of a publication is to be instructive and effective.** Our work achieves this by introducing a principled framework that makes it possible to **eliminate** the learning rate decay phase entirely. This contribution is valuable precisely because it provides a unified and practical solution to a pervasive problem in model training.
>
> Our framework enables the decoupling the training process from a pre-defined decay strategy. Instead of being locked into a specific schedule (e.g., linear or cosine), practitioners can now emulate the effect of various strategies after training is complete. This not only makes agile pre-training possible but also provides a unified, flexible, and highly effective alternative to the rigid decay paradigm. **This represents a paradigm shift, not repackaging.** We believe this principled and practical contribution is highly valuable to the community.
>
> While the primary aim of our work is to be instructive and effective, we did in fact arrive at a surprising and significant finding: our experiments reveal that the most widely-used EMA methods are not optimal, and instead, mean merging and 1-sqrt merging strategy delivers the better performance—an insight that has been largely overlooked in prior literature. We believe this empirical discovery effectively opens a new direction for exploring a wider space of merging strategies, which will yield further "surprising" and impactful results as the reviewer anticipates.

---

> ### Author Response · Authors · 2025-11-20
> **Rebuttal by Authors (part-2/3)**
>
> > **Weakness 3: From the experiments it is not clear why or if merging along the entire training trajectory would hurt.**
>
> **Response:** While an online running average across the entire training trajectory is simpler to implement, our offline merging approach provides two critical advantages: it achieves **better final performance** and offers **far greater flexibility at a lower cost**.
>
> 1. **Risk of Suboptimal Performance:**
> An online approach, like EMA, forces a commitment to a single, irreversible averaging scheme from the outset. This is a significant drawback, as the optimal merging strategy is often unknown a priori and varies with the training setup. For example, our experiments conclusively show that EMA is significantly outperformed by offline methods like mean and 1-sqrt merging. Correcting a poor choice would necessitate cost-prohibitive retraining.
>
> 1. **Lack of Flexibility and Cost-Effectiveness:**
> The core value of our offline, checkpoint-based approach is its exploratory flexibility. We can test dozens of virtual decay schedules (different merge algorithms and durations) on a single set of saved checkpoints. This is far more cost-effective, as storage is significantly cheaper than re-computation, and it provides the flexibility needed to find the optimal merge configuration.
>
> In summary, the simplicity of online averaging comes at the steep price of lower performance and zero flexibility. Our method provides the control and exploratory power crucial for optimizing large-scale pre-trained models.
>
> &nbsp;
> ---
>
> >**Question 1: Section 5.2: What you are calling model merging in this paper is more commonly termed as Stochastic Weight Averaging (SWA).**
>
> **Response:** Our choice of the term "model merging" was motivated by its recent and broader usage in the literature [5, 6] to describe a category of techniques that includes averaging checkpoints from a single training run. To enhance clarity, we will revise the manuscript to explicitly introduce SWA, and discuss its relationship to the broader concept of model merging.
>
> &nbsp;
>
> >**Question 2: WSD is not exactly schedule-free as it has been found that optimal LR depends on the decay start time. By extension WSM is also not schedule-free. Please fix this claim, or at least add a caveat and discussion.**
>
> **Response:** We use the term "**decay-free**" specifically rather than "**schedule-free**" in the our paper title and context of the training process itself.
>
> &nbsp;
>
> >**Question 3: Please remove L115.**
>
> **Response:** Thank you for this sharp observation. We will remove the line and add a discussion to the limitations section clarifying that while WSM still need / will benefit from LR tuning.
>
> &nbsp;
>
> >**Question 4: L257 corresponds to a linear decay LR schedule.**
>
> **Response:** Thanks for your suggestion. We will clarify that this corresponds to a linear decay schedule.
>
> &nbsp;
>
> >**Question 5: What is the intuition for why WSM is better than WSD?**
>
> **Response:** WSM is superior to WSD for two reasons.
>
> 1. **Intuition and Empirical Analysis: Flatter Minima and Better Generalization.**
> The primary intuition is that the WSM's averaging process guides the model towards a wider, flatter minimum in the loss landscape, which is strongly correlated with better generalization. In contrast, a standard WSD trajectory can converge to a sharper, less robust minimum. We provide direct empirical evidence for this in our loss landscape visualization in Figure 9 of Appendix.
>
> 2. **Practical Advantage: Decoupled Exploration and Cost-Efficiency.**
> Beyond performance, WSM's most significant practical advantage is its flexibility and cost-effectiveness. WSM decouples training from schedule selection. From a single set of saved checkpoints, we can efficiently explore dozens of virtual decay schedules post hoc. This dramatically reduces the computational cost and risk associated with finding the optimal schedule.
>
> &nbsp;
>
> >**Question 6: Why is there a systematic dip in all plots towards the end in Figure 4?**
>
> **Response:** This dip is attributable to normal stochastic fluctuations during pretraining. **The final saved checkpoint in that particular run happened to have slightly lower evaluation performance.** Including it in the merge window results in a minor, reflecting the inherent noise of the training process.

---

> ### Author Response · Authors · 2025-11-20
> **Rebuttal by Authors (part-3/3)**
>
> >**Question 7: L348: "this finding reinforces the theoretical connection"... I believe you need Figure 5a style curves to rigorously establish this.**
>
> **Response:** Our argument for the theoretical connection is grounded in the consistency of performance rankings between the WSD and WSM, i.e., what works best for WSD also works best for WSM. As shown in our experiments (Appendix E):
>
> * WSD performance ranking (**decay**): 1-sqrt > linear > exponential (EMA)
> * WSM performance ranking (**merging**): 1-sqrt > linear > EMA
>
> This identical ranking demonstrates that WSM reliably reproduces the relative effectiveness of different decay schedules. We will clarify that this corresponds in the revised manuscript.
>
> &nbsp;
>
> >**Question 8: L429: Is "language modeling loss" column the train loss or test loss?**
>
> **Response:** This refers to the test loss. We will clarify this in the table caption.
>
> &nbsp;
>
> >**Question 9: L483: The theory is not exactly optimizer-agnostic with a more careful analysis.**
>
> **Response:** Thanks for your suggestion. We will revise the claim from "optimizer-agnostic" to the more precise "the method is applicable across different optimizers".
>
> More importantly, to empirically validate this claim, we have conducted a **new experiment comparing WSM and WSD with an additional, representative optimizer: Muon**. We performed 100B token enhancement training on 2T checkpoints, using Muon optimizers, evaluated on the same benchmark as in the main paper.  The results, shown below, demonstrate that WSM achieves significant gains over WSD regardless of the Muon optimizer.
>
> Table 1: WSD vs. WSM performance (%) with Muon optimizer on 100B-token (from 2T checkpoints).
> |     |   WSD |   WSM |
> |:-----|------:|------:|
> | General Knowledge/ Reasoning     | 63.37 | **63.95** |
> | Professional Knowledge     | **45.09** | 45.04 |
> | Language Understanding     | 75.50 | **76.95** |
> | Math     | 55.01 | **56.85** |
> | Code     | 47.10 | **47.30** |
> | Overall Average     | 56.10 | **56.74** |
>
> We will integrate these findings and the revised claim into the manuscript.
>
>
> &nbsp;
>
> ---
>
> **References:**
>
> [1] Straight to zero: Why linearly decaying the learning rate to zero works best for LLMs. ICLR 2025.
>
> [2] How to set AdamW's weight decay as you scale model and dataset size. ICML 2025.
>
> [3] The Llama 3 Herd of Models.
>
> [4] Post-Hoc Reversal: Are We Selecting Models Prematurely? NeurIPS 2024.
>
> [5] Model Merging in Pre-training of Large Language Models. (https://arxiv.org/abs/2505.12082)
>
> [6] Checkpoint merging via bayesian optimization in llm pretraining. (https://arxiv.org/abs/2403.19390)

---

### Official Review · Reviewer_kALL · 2025-11-01

**Soundness:** 3
**Presentation:** 4
**Contribution:** 4
**Rating:** 10
**Confidence:** 4

**Summary:**

The authors present a theoretically grounded approach to model merging, which they suggest could fully replace LR decay schedules for WSD. This would be great! The basic idea is that, since every theta_k is a weighted combination of the gradients so far, you can choose a particular weighted average of the theta_k to emulate any decay schedule you please. They show the algebra to justify this and then a bunch of experiments at a more-than-reasonable scale that very convincingly show that you can, on tasks, do every bit as well as a proper decay (if not better).

**Strengths:**

Overall I really like this paper! It's well written and argued, and the empirical results are quite strong. In terms of significance, this has a non-trivial chance of becoming the standard way people produce final model checkpoints.

The theoretical derivation of the equivalence is simple in a wonderful way (though I worry a bit specious). It's of course just algebra, but I have no problem with that. The intuition is the same one that motivates EMA and SWA etc.

 The empirical results are very strong, with across the board accuracy gains relative to EMA (another decay-free strategy) and even over WSD itself.

They also do a fairly good job defending the proposed equivalence: averaging schemes derived from stronger decay schedules (1-sqrt, linear) outperform those derived from weaker schedules (exponential decay--> EMA); and they show little to no benefit of combining a real decay with averaging, strengthening the claim that they are in some sense equivalent.

**Weaknesses:**

I have two categories of concerns:

### Experiments don't really demonstrate connection between theory and practice.

My biggest concern is that, algebra-aside, the theory is a bit specious. The derivation assumes that gradients would be ~the same in a decay versus non-decay setup, and that can't be true. So, really I think this is "decay-inspired averaging" or "empirical substitute" more than an equivalent/replacement.

To that end, there aren't any experiments directly comparing the supposedly equivalent decay schedules with merges, except for the linear decay schedule, and even that experiment isn't fully convincing. I get that you can't afford to do this for a model trained to 10T tokens, but I would expect it wouldn't be too hard to do this for a smallish model?

For instance, if you were to really believe the theory strongly enough, you could, after warmup, keep the LR hot the entire time and recover a cosine decay via appropriate weighting. Right? Do experiments support this? If not, where does the theory/reality connection break down?

I would expect that the losses wouldn't be that similar, because this approach assumes the gradients you get are similar enough in a decay vs non-decay setup that the averaging is ~equivalent to decay.

I'd expect these to show up in loss plots rather than accuracy plots. I note that loss plots are conspicuously absent from the paper, which to me suggests that WSM isn't so much "equivalent to WSD" but more "just as good as" (which is more than good enough, given the advantages!). I just think we should see these results and adjust the positioning of the paper as appropriate.

### Can't avoid data scheduling

Another, smaller weakness is that, in the age of "mid-training", you still have to decide when to start including higher quality data. In typical WSD setups, that is also when you start the decay. So you don't really have the nice any-time property that you might want from a truly decay-free world. I'm not sure how to work around this. It just (slightly) diminishes the promise of this approach.

**Questions:**

As noted above, it would be nice if you were able to conduct some preliminary experiments comparing other decay schedules at a modest scale 150M/3B or something. (A different scale is fine too.)

Again, if you fully believe the theory, you should be able to approximately recover a 100% cosine schedule by starting your averaging/merging from the end of warmup. I think?

I recognize that your pipeline isn't set up for this (and I think it was a good choice for your experiments), and I don't expect you to run a large experiment here, but in my experience with WSD with EMA merging, we just update the merged theta at every step. This is obviously a lot more merges, but it's cheap and online. One could also do this with mean averaging. And Figure 4 indicates that it's definitely better to merge more often! how does maintaining a running average every step compare?

---

> ### Author Response · Authors · 2025-11-20
>
> > **Weakness 1: Experiments don't really demonstrate connection between theory and practice.**
>
> **Response:**  We completely agree that our theory provides a powerful framework for guiding practice rather than a strict, point-wise mathematical equivalence. Our primary goal is to demonstrate that checkpoint merging can serve as a principled and effective simulation of the LR decay process, thus fully replacing LR decay with checkpoint merging. **Our argument for the theoretical connection is grounded in the consistency of performance rankings between the WSD and WSM, i.e., what works best for WSD also works best for WSM. As shown in our experiments (Appendix E):**
>
> * WSD performance ranking (**decay**): 1-sqrt > linear > exponential (EMA)
> * WSM performance ranking (**merging**): 1-sqrt > linear > EMA
>
> **This identical ranking demonstrates that WSM reliably reproduces the relative effectiveness of different decay schedules.**
>
> &nbsp;
>
> > **Weakness 2: Can't avoid data scheduling**
>
> **Response:** Yes. Our work focuses on replacing the learning rate decay schedule, not the data schedule. Our method is designed to make the learning rate decay a post-hoc decision, providing significant flexibility. The scope of our work is to decouple the learning rate decay from the training process, not to address data scheduling. Therefore, **WSM is compatible with any data scheduling strategy and does not affect the effectiveness of the data scheduling.**
>
> &nbsp;
>
> >**Question 1: Replicating other decay schedules (e.g., 100% cosine).**
>
> **Response:** We sincerely thank the reviewer for the suggestion. For a discussion of alternative decay methods, please refer to our response to Weakness 1.
>
> Averaging weights over the entire training trajectory would incorporate checkpoints from the highly volatile early stages of pre-training. Our empirical observations show that these early-stage fluctuations can introduce significant noise and degrade the quality of the final merged model.
>
> &nbsp;
>
> >**Question 2: Comparison to an "online" running average.**
>
> **Response:** Thank you for this follow-up question. **An online running average, while memory-efficient, sacrifices the core flexibility advantages of WSM and is demonstrably suboptimal.**
> 1. **It Forfeits Flexibility:** The primary value of WSM is the ability to **explore dozens of different virtual decay schedules from a single set of saved checkpoints** without any retraining. An online average **commits the training to a single, irreversible trajectory**. If that choice proves suboptimal, the only recourse is expensive retraining.
> 2. **It is Restrictive:** A recursive, online running average is similar to EMA. Our experiments clearly show that EMA is an inferior merging strategy compared to non-uniform schemes like 1-sqrt. An online average is fundamentally incapable of simulating these more effective schedules, making it a less powerful approach.
>
> Therefore, an online average is not a suitable replacement for WSM's flexible and effective offline methodology.

---

### Official Review · Reviewer_hHGj · 2025-11-01

**Soundness:** 3
**Presentation:** 4
**Contribution:** 3
**Rating:** 10
**Confidence:** 4

**Summary:**

This paper proposes an alternative approach to WSD (Warmup-Stable-Decay) learning rate schedule. It proposes to use a constant learning rate and then apply weight merging on the saved checkpoints. The authors establish an informal correspondence between the two methods and show empirically some of the weight merging schedules can result in higher downstream performance than the WSD method.

**Strengths:**

1. The experiment in this paper is large-scale and detailed. The models are evaluated over a range of hard benchmarks to show the improvement of the methods.

2. The proposed method is conceptually clean and can be useful in many settings.

3. The theoretical justification of the method, while not rigorous, provides intuitions in guiding the design of the algorithm.

**Weaknesses:**

1. In Table 5, the authors show that WSM has a higher ending loss compared to WSD. This seems to be counterintuitive, especially given the higher downstream accuracy. This also brings questions regarding the validity of the correspondence discussed in Section 3.1.

2. Regarding continual pretraining, the authors propose to continual pretrain from the constant learning rate checkpoints before weight merging. It is unclear how this will compare with re-warming up the decay checkpoint in a standard WSD setting.

**Questions:**

1. The authors mention sensitivity regarding the weight merging schedules. Is there similar sensitivity in the decay schedule in WSD? Also, is the ranking between the weight merging schedules consistent with the corresponding learning rate schedule?

2. Do the authors observe differences in the ranking of weight merging schedules when choosing different downstream evaluation tasks?

---

> ### Author Response · Authors · 2025-11-20
>
> We sincerely thank the reviewer for the suggestion, and list our responses below.
>
> ---
>
> > **Weakness 1: WSM has a higher pre-training loss but better downstream performance.**
>
> **Response:**  Thank you for this insightful point. **The higher pre-training loss of WSM is not counterintuitive: weight merging can move the model away from sharp, less robust optima toward flatter regions that generalize better, yielding stronger downstream performance (see Fig. 9, Appendix G).** By contrast, WSD’s lower loss likely reflects overfitting to the sharp local optima [1].
>
> Furthermore, loss does not always correlate directly with accuracy. For instance, mixing instruction-following data into the training set does not substantially reduce loss, yet it can yield significant performance improvements. Thus, we believe the observed relationship between loss and accuracy is reasonable to some extent.
>
> &nbsp;
>
> > **Weakness 2: It is unclear how this will compare with re-warming up the decay checkpoint in a standard WSD setting.**
>
> **Response:** In our experiments, the WSD baseline **already employs the standard re-warming–up strategy for continual pre-training**. We will clarify this in the revision.
>
> &nbsp;
>
> > **Question 1: Is there similar sensitivity in the decay schedule in WSD? Also, is the ranking between the weight merging schedules consistent with the corresponding learning rate schedule?**
>
> **Response:** Yes, decay schedules in WSD exhibit similar sensitivity, and the performance rankings of schedules between WSM and WSD are highly consistent, supporting the theoretical connection. Our experiments in **Appendix E of our inital manuscript** establish a clear performance ranking for WSD schedules, and this ranking is consistent with the performance ranking of corresponding WSM schedules:
>
> * WSD performance ranking (**decay**): 1-sqrt > linear > exponential (EMA)
> * WSM performance ranking (**merging**): 1-sqrt > linear > EMA
>
> This strong consistency provides direct empirical validation for the theoretical connection between LR decay and checkpoint merging that our work establishes.
>
> &nbsp;
>
> > **Question 2: Do the authors observe differences in the ranking of weight merging schedules when choosing different downstream evaluation tasks?**
>
> **Response:** We observed that the rankings of merging schedules are **consistent across more than half of the task categories**, with occasional inconsistencies likely stemming from the stochasticity of training.
>
> The performance ranking of the schedules is broadly consistent across the majority of downstream task categories, including general knowledge, professional knowledge, and code. While minor variations exist in some categories, which we attribute to standard training stochasticity, the overall trend holds. This consistency reinforces that the benefits of certain merging schedules are not task-specific but reflect a more fundamental improvement in the model's generalized capabilities.
>
> &nbsp;
>
> ---
>
> **References:**
>
> [1] Izmailov, Pavel, et al. "Averaging weights leads to wider optima and better generalization." arXiv preprint arXiv:1803.05407 (2018).

---

> > ### Comment · Reviewer_hHGj · 2025-11-26
> >
> > Thank you for the detailed rebuttal. I think the instruction following mixing is not a proper analogy as we are discussing loss comparison on the same dataset. This being said, it is right that lower loss may not be equivalent to better downstream performance so I agree that this is not an important issue. I will keep my score.

---

### Author Response · Authors · 2025-11-24
**General Response and PDF Revision Summary**

We would like to thank all the reviewers for their valuable feedback. We are glad that reviewers found our paper to be **well-written and clearly argued** (Reviewer kALL), with a proposed method that is **conceptually clean, simple, and practical** (Reviewer hHGj, fGtp). Reviewers also highlighted our **comprehensive, large-scale, and strong empirical experiments** (Reviewer hHGj, kALL, fGtp, 852L), while recognizing the **insightful theoretical connection** that provides valuable intuition (Reviewer hHGj, kALL, 852L). Finally, we are encouraged that our work is seen as having **significant potential impact**, with one reviewer noting it could become a new standard practice (Reviewer kALL).


&nbsp;


In response to the reviewers' insightful feedback, we have revised the paper accordingly. The main changes are as follows:

1. **Clarification on Theoretical Connection**  (for Reviewer hHGj, kALL and fGtp):

   In Section 4.3.2, we explicitly strengthened the argument for our theoretical framework. We highlighted the consistency of performance rankings between WSD and WSM to empirically reinforce the connection between theory and practice.
1. **Clarification of Contributions and Terminology (SWA)** (for Reviewer fGtp):

   To clarify terminology, we now explicitly define Stochastic Weight Averaging (SWA) in Section 5.2. To distinguish our work from prior approaches, we enriched the related work section and added a section in Appendix C that contrasts our  work with that of prior representative studies (e.g., PMA and ERNIE-EMA).
1. **Discussion on Online Methods** (for Reviewer kALL):

   In Appendix C, we added a detailed discussion comparing our approach with online methods. We explicitly highlight the advantages of WSM in terms of flexibility and expressiveness.
1. **New Experiments with Muon Optimizer** (for Reviewer fGtp):

   We have added experimental results using the Muon optimizer in Appendix D. These results demonstrate that our method remains effective across diverse optimization algorithms, validating its broad applicability.
1. **Other Improvements for Clarity** (for Reviewer hHGj, kALL, 852L and fGtp):

   We have added details and improved the phrasing in Section 2 (Preliminary), Section 4.1 (Experiment Setup), and Section 6 (Conclusion) to enhance the overall clarity and precision of the manuscript.

Once again, we thank all the reviewers for their constructive comments, which have been invaluable in strengthening the manuscript.

---

### Public Comment · ~Niccolò_Ajroldi1 · 2025-11-24
**Experimental Setup and Theoretical Considerations**

I would like to raise two discussion points.

**1. Why do the authors use such a short cooldown in WSD?**

Prior works on WSD [1,2] recommend using decay durations of at least 10-20% of total tokens, and several model families adopt similar strategies ([SmoLM3](https://huggingface.co/blog/smollm3): 10%, [Apertus](https://arxiv.org/abs/2509.14233): 10%, [Kimi-K2](https://arxiv.org/pdf/2507.20534): 35%). Here, however, the LR is decayed for only 400B out of 10.6T tokens, which amounts to a **cooldown duration of 3.8%**. This unusually short cooldown may explain why WSD underperforms WSM.

As the goal is to compare WSM and WSD, the WSD baseline should be carefully **tuned**, or at least aligned with common values. While WSM outperforms WSD *under the paper’s specific setup*, this choice may limit the scope of the result.

Additionally, although downstream tasks are informative, no **loss curves** are reported, making it difficult to assess any claim of equivalence and reconcile results with previous evidence [1].

**2. What is the primary theoretical contribution?**

The paper aims to _“provide a principled approach to convert any LR decay method into a theoretically approximate model averaging implementation”_. However, the proposed argument does not formally support such correspondence.

The theorem itself is correct: averaging SGD iterates yields a weighted sum of the gradients computed *along that same trajectory*. However, this does not imply equivalence between running SGD *with a decaying step-size* and averaging iterates obtained from *constant step-size* SGD, since the two procedures follow different trajectories and produce different gradients, making the theorem unsuitable for practical use. Overall, the paper highlights a *connection* between LR decay and averaging, rather than an *equivalence*.

On the other hand, such *connection* between averaging and LR decay has long been recognized in the optimization literature [3]. In particular, [4] has recently shown the *equivalence* of LR decay and averaging under a noisy quadratic model, deriving **equivalent schedules** and testing them empirically.

**References**

[1] Hägele, A. et al. (2024). Scaling Laws and Compute-Optimal Training Beyond Fixed Training Durations. arXiv:2405.18392.\
[2] [HuggingFace training playbook](https://huggingface.co/spaces/HuggingFaceTB/smol-training-playbook#rules-of-engagement) \
[3] Bottou, L., Curtis, F. E., & Nocedal, J. (2018). Optimization Methods for Large-Scale Machine Learning. SIAM Review, 60(2), 223-311. \
[4] Sandler, M., et al. (2023). Training trajectories, mini-batch losses and the curious role of the learning rate. arXiv:2301.02312.

Thank you for considering this comment, I am glad to discuss further.

---

> ### Author Response · Authors · 2025-11-26
>
> We sincerely thank the reviewer for their insightful comments.
>
> &nbsp;
>
> > **Q1: Why is the WSD cooldown so short in the main experiment?**
>
> Our choice was deliberate and motivated by practical considerations, and we also conducted additional experiments to validate our claims under the conditions suggested.
>
> * **Shorter cooldown reflects practical data constraints in LLM pretraining.**
>
>   In realistic LLM pretraining workflows, cooldown is typically performed on the highest-quality data available. At trillion-token scales, assembling 10–35% of the total training tokens at comparable quality is often unfeasible for most developers. For example, if a base model has already consumed 40T tokens (e.g., Llama 4 scale), obtaining an additional 4T high-quality tokens solely for cooldown is beyond reach for most practitioners. Our training setup in the main experiment faced a similar limitation on the volume of our enhancement data. This choice reflects production constraints rather than an under-tuned baseline.
>
> * **Additional experiments with longer cooldown (>10%) show WSM still wins.**
>
>   To directly address your concern about the fairness of the comparison, we conducted a new controlled experiment. Starting from a 500B-token checkpoint, we continued training for an additional 200B tokens. In this setup, the WSD baseline employed a decay schedule over the entire 200B tokens, resulting in a cooldown ratio of **~28.6%** (200B / 700B total), which comfortably satisfies the ≥10% recommendation. The results below show that WSM still outperforms WSD on the overall average and on most sub-categories. We will provide detailed plot of training dynamic in the revised manuscript.
>
>     | Metric                      | WSM | WSD |
>     | :-------------------------- | :-----------: | :-----------: |
>     | **Overall Average**         |     **51.43**     |     50.57     |
>     | General Knowledge/Reasoning |     56.47     |     **56.60**     |
>     | Language Understanding      |     **67.04**     |     66.44     |
>     | Professional Knowledge      |     **37.46**     |     37.25     |
>     | Math                        |     **51.97**     |     50.05     |
>     | Code                        |     **45.84**     |     44.30     |
>
> * **On loss curves vs. downstream performance.**
>
>   We acknowledge that loss curves were not included in the initial submission. However, we wish to emphasize that in the context of LLMs, validation loss and downstream benchmark capabilities may diverge. As our primary goal is to improve the model's practical abilities, we prioritized benchmark evaluation as the most direct measure of success. Nonetheless, we will analyse loss curves in the furture and discuss their relationship to downstream results.
>
>
> This finding reinforces our core message: WSM provides valuable flexibility for LLM pretraining and demonstrates the unique advantages of checkpoint merging over LR decay in the context of large-scale pre-training with massive compute and evolving data distributions.
>
> &nbsp;
>
> > **Q2: What is the primary theoretical contribution?**
>
> We would like to clarify the nature and scope of our contribution.
>
> * **A principled approximation for practice.**
>
>    We fully agree that our method establishes a "**approximate**" relationship, not a formal mathematical equivalence, as the optimization trajectories differ. This is why we chose our wording carefully. However, we respectfully disagree with the conclusion that this makes our theorem "unsuitable for practical use." As Reviewer-kALL noted, our approach is best understood as a "**decay-inspired averaging**" framework. Its value lies in providing a principled, effective guide for practice. The empirical superiority of our method over a well-tuned WSD baseline demonstrates its immense practical utility.
>
> * **Bridging theory and large-scale practice.**
>
>    Our primary contribution is not rediscovering the link between averaging and decay or analyzing why standard averages like EMA/SWA work. Our framework serves a different purpose: it enables the design of novel and superior averaging strategies for LLM pretraining. For instance, we use our framework to implement a 1-sqrt merging scheme, empirically proving it outperforms the standard EMA baseline. We operate this principle into WSM. Crucially, we validate our approach in the complex, large-scale regime, bridging a critical gap between established theory and state-of-the-art practice.
>
>
> Thank you again for your insightful feedback. We will revise the manuscript to include the new experimental results and the above discussion. We believe your comments have been invaluable in helping us improve our work.

---

> ### Public Comment · ~Niccolò_Ajroldi1 · 2025-11-26
> **Comment**
>
> Thank you for the answer and additional experiments!
>
> I would like to note that when high-quality data are scarce, the cooldown is typically **not shorten**. Instead, practitioners generally rely on *"synthetic data generation strategy to increase token utility"* ([Kimi-K2](https://arxiv.org/pdf/2507.20534), 35% decay, 15T tokens), or *"upsample high quality data"* ([SmoLM3](https://huggingface.co/blog/smollm3), 10% decay, 1.1T tokens) to allow for a **longer**, better performing cooldown duration.
>
> In addition, it would be helpful to include details and ablations regarding the **tuning** of the WSD baseline, as careful tuning is a crucial part of evaluating an optimization method [1]. Since previous works [2,3] have suggested that averaging over a constant learning rate does **not match** LR-decay performance, further evidence, ideally across a broader range of settings, would help support a different conclusion.
>
> Thank you again for the discussion.
>
> [1] Dahl, G. E., et al. (2023). Benchmarking Neural Network Training Algorithms. arXiv:2306.07179. \
> [2] Hägele, A. et al. (2024). Scaling Laws and Compute-Optimal Training Beyond Fixed Training Durations. *Advances in Neural Information Processing Systems*. \
> [3] Ajroldi, N., et al. (2025). When, Where and Why to Average Weights? _Proceedings of the 42nd International Conference on Machine Learning_.

---

> > ### Author Response · Authors · 2025-11-27
> >
> > Thank you for your reply. We would like to offer further clarification and a response to the points you raised.
> >
> > &nbsp;
> >
> > **1) Regarding cooldown duration**
> >
> > We agree that extending the training in the cooldown phase using methods like "synthetic data generation" or "upsampling high-quality data" is an effective and common strategy. It is precisely to address the potential impact of cooldown duration that we conducted the additional controlled experiment described in our previous response. In this experiment, we employed a cooldown ratio of **~28.6%**, which comfortably satisfies the standard recommendation. **The experiment results showed that WSM still outperformed the WSD baseline with a cooldown ratio greater than 10%**. We believe this empirical evidence confirms that the advantage of WSM is robust and is not merely an artifact of the specific cooldown length used in the main experiments.
> >
> > &nbsp;
> >
> > **2) Regarding WSD baseline tuning and comparison with previous work**
> >
> > We want to assure you that **the baseline was tuned carefully**. In our preliminary experiments (Appendix F), we evaluated various decay schedules and selected the 1-sqrt schedule as it proved to be the strongest baseline. This choice is also supported by other studies [2]. We acknowledge the apparent discrepancy between our conclusion and that of previous studies [2,3]. We believe this stems from two key differences:
> >
> > * **Difference in Observation Metric:**
> >
> >   **Our conclusions are drawn from the model's comprehensive performance across a wide range of downstream benchmarks, whereas previous studies often focused more on the loss value or a specific metric.** Interestingly, if we were to focus solely on the loss value or a subset of benchmarks, our results would **align** with their findings. For example, the loss of our merged checkpoint is indeed slightly higher than that of the final decayed checkpoint (see Section 4.3.5). As another example, in terms of comprehensive knowledge/reasoning ability (including tests such as ARC, AGIEval, OpenBookQA, BBH, WorldSense, PIQA and hellaswag) of the aforementioned experiments, the performance of WSM does not match that of WSD. However, it is the superior overall capability demonstrated by WSM on a broader set of benchmarks that led us to our final conclusion.
> >
> > * **Difference in Training Practice:**
> >
> >   **Our main experiments simulated a common practice in LLM pre-training: during the LR decay phase, cooldown data is simultaneously introduced. In contrast, previous studies may have focused more on the optimization method itself, disregarding potential data shifts during the LLM pre-training.** In the practice of LLM pre-training, WSM with a constant learning rate allows the model to better "digest" the cooldown data, thereby achieving better performance. This is also consistent with findings from some subsequent work [4].
> >
> > Furthermore, rather than developing a general optimization algorithm for broad machine learning problems, this work aims to present better practices **tailored to the real-world scenarios of LLM pretraining**. It is for this reason that we conscientiously use the wording "for LLM pretraining" in our title and main text.
> >
> > &nbsp;
> >
> > Thank you again for your insightful feedback. We will revise the manuscript to further discussion regarding the similarities and differences with the previous studies [2,3,4].
> >
> > &nbsp;
> >
> > Reference:
> >
> > [1] Dahl, G. E., et al. (2023). Benchmarking Neural Network Training Algorithms. arXiv:2306.07179.
> >
> > [2] Hägele, A. et al. (2024). Scaling Laws and Compute-Optimal Training Beyond Fixed Training Durations. Advances in Neural Information Processing Systems.
> >
> > [3] Ajroldi, N., et al. (2025). When, Where and Why to Average Weights? Proceedings of the 42nd International Conference on Machine Learning.
> >
> > [4] Luo, K., et al. (2025) How Learning Rate Decay Wastes Your Best Data in Curriculum-Based LLM Pretraining. [arxiv 2511.18903](https://arxiv.org/abs/2511.18903)

---

### Comment · Area_Chair_Nhkz · 2025-11-28

Dear Reviewers,

The discussion phase is now underway, and the authors have finished uploading their responses to reviewers. If you haven't already, please carefully review the authors' responses to understand their perspectives. Engage in thoughtful, constructive discussions with authors, sharing your thoughts and seeking clarifications. Please also update your review or rating if necessary.

It is noted in the guideline that reviewers can leave comments visible to authors **until Dec 2 11:59pm AoE**. Your active participation and contribution to the ongoing discussion are highly encouraged. Thank you very much for your contribution to ICLR.

Best regards,

AC

---

### Author Response · Authors · 2025-12-04
**Summary of Our Responses and Revisions during the Rebuttal Period**

Dear Area Chairs,

We sincerely thank you for your time and effort in handling our paper.

Our work introduces the **Warmup-Stable-Merge (WSM) LR Schedule**, a decay-free pre-training paradigm that replaces the LR decay of the Warmup-Stable-Decay (WSD) schedule with offline checkpoint merging, **significantly enhancing both flexibility and performance in LLM pre-training**.

---

# 1. Reviewer Feedback
The review process was highly constructive, resulting in **two "strong accept"** ratings and allowing us to significantly improve the paper by addressing all raised concerns.

*   **`Reviewer hHGj (10, keeping score)`** found the paper strong and well-presented, highlighting its conceptual clarity, large-scale experiments, and useful theoretical intuitions. After reviewing our detailed responses, they **decided to keep the strong accept rating**.
*   **`Reviewer kALL (10, no further discussion)`** highlighted the work's potential to become a "**standard way people produce final model checkpoints**" and praised the "large-scale and detailed" experiments. Our rebuttal addressed their insightful questions regarding the method's properties and the connection between theory and practice.
*   **`Reviewer 852L (6, no further discussion)`** acknowledged the paper's formal connections and thorough analysis. We addressed their concerns about computational overhead by clarifying our method's significant **cost-effectiveness and risk-reduction benefits**.
*   **`Reviewer fGtp (2, no further discussion)`** considered the method simple and effective but raised concerns about the work's novelty, which we believe stemmed from a **misunderstanding** of its core contributions. Our rebuttal clarified the work's paradigm-shifting contributions by distinguishing it from prior methods and providing new experimental evidence. Unfortunately, no further discussion was possible following our clarification.

---

# 2. Our Strengths
We are grateful for the reviewers' positive feedback and summarize their key acknowledgements below:

*   **Novelty and Impact:** Reviewers highlighted the novelty of our approach, noting the "theoretically grounded approach for model merging" (`kALL`) with an "insightful" formal connection to LR decay (`852L`). They highlighted its potential to become "the standard way" for producing final models (`kALL`).
*   **Large-Scale and Comprehensive Experiments:** All reviewers praised our experiments **(16B MoE, over 10T tokens)** as "large-scale and detailed" (`hHGj`), "very convincingly" demonstrating the method's effectiveness at a "more-than-reasonable scale" (`kALL`), and "comprehensive" (`fGtp`).
*   **Practicality and Soundness:** The method was described as "conceptually clean and useful" (`hHGj`) with a "strong defense" of its core claims (`kALL`). Furthermore, the paper was commended for its "excellent" presentation and for being "well written and argued" (`hHGj`, `kALL`).

---

# 3. Main Responses and Revisions
In response to their feedback, we have revised the paper accordingly. The main changes are as follows:

*   **Expanded Discussions:**
    *   **Clarification on Theoretical Connection** (for Reviewers `hHGj`, `kALL`, and `fGtp`):
        In Section 4.3.2, we explicitly strengthened the argument for our theoretical framework. We highlighted the consistency of performance rankings between WSD and WSM to empirically reinforce the connection between theory and practice.
    *   **Clarification of Contributions and Terminology** (for Reviewer `fGtp` and Public Comment):
        To clarify terminology, we now explicitly define Stochastic Weight Averaging (SWA) in Section 5.2. To distinguish our work from prior approaches, we enriched the related work section and added a section in Appendix C that contrasts our work with that of prior representative studies (e.g., ERNIE-EMA and PMA).
    *   **Discussion on Other Related Works** (for Reviewer `kALL`):
        In Appendix C, we added a detailed discussion comparing our approach with online methods. We explicitly highlight the advantages of WSM in terms of flexibility and expressiveness.
*   **New Experiments with the Muon Optimizer** (for Reviewer `fGtp`):
    We have added experimental results using the Muon optimizer in Appendix D. These results demonstrate that our method remains effective across diverse optimization algorithms, validating its broad applicability.
*   **Other Improvements for Clarity** (for Reviewers `hHGj`, `kALL`, `852L`, and `fGtp`):
    We have added details and improved the phrasing in Section 2 (Preliminaries), Section 4.1 (Experiment Setup), and Section 6 (Conclusion) to enhance the overall clarity and precision of the manuscript.

---

Once again, we thank all the reviewers for their constructive comments. **All changes have been incorporated into the revised version, with modifications clearly highlighted.**

Thank you once again for your valuable time and consideration.

Best Regards,

Authors

---

### Meta-Review · Area_Chair_VWq3 · 2025-12-05

**Summary:**

The authors propose a way to simulate learning rate decay methods via model merging. LR decay methods require one to decide ahead of time when to start decaying. The proposed method instead uses weight averaging with previous checkpoints, showing that this is, under some  assumptions, equivalent to LR decay. This seems like a novel use of model merging, and impactful for practitioners as it eliminates the need to guess when to start decaying.

I would like to note that the authors missed some older but very relevant work: lookahead optimizers (Zhang et al). They should explicitly compare against this work, and discuss the connections.

**Reviewer Concerns:**

Reviewers pointed out that the theory might not hold for all optimizers, and the authors acknowledged. Also, the reviewers noted that the proposed link between the merging method and lr decay is only approximate (gradients would have been different under a different learning rate schedule). The authors seemed to acknowledge this too. Another reviewer pointed out similarity to SWA, and the authors responded explaining how SWA is a special case of their method.
Finally, another concern expressed by the reviewers was storage costs, but the authors noted that one stores previous checkpoints anyway in large scale training, thus adding little in terms of additional cost.

**Reviewer Scores:**

Some of the reviewers did engage. I would expect Reviewer 852L to raise their score: main concern was regarding computational and storage overhead, and a lack of baselines like SWA. The authors provided a strong rebuttal regarding storage, noting that saving checkpoints is already mandatory for fault recovery in large-scale training, meaning their method incurs zero additional storage cost beyond standard practice. They also clarified that "mean merging" is essentially SWA, addressing the missing baseline concern.

---

### Decision · Program_Chairs · 2026-01-26

Accept (Oral)